# Coinfection frequency in water flea populations is a mere reflection of parasite diversity
Snir Halle [1] ✉, Ofir Hirshberg[1], Florent Manzi[2], Justyna Wolinska[2,3] & Frida Ben-Ami [1]

In nature, parasite species often coinfect the same host. Yet, it is not clear what drives the natural dynamics of coinfection prevalence. The prevalence of coinfections might be affected by interactions among coinfecting species, or simply derive from parasite diversity. Identifying the relative impact of these parameters is crucial for understanding patterns of coinfections. We studied the occurrence and likelihood of coinfections in natural populations of water fleas (*Daphnia magna*). Coinfection prevalence was within the bounds expected by chance and parasite diversity had a strong positive effect on the likelihood of coinfections. Additionally, coinfection prevalence increased over the season and became as common as a single infection. Our results demonstrate how patterns of coinfection, and particularly their temporal variation, are affected by overlapping epidemics of different parasites. We suggest that monitoring parasite diversity can help predict where and when coinfection prevalence will be high, potentially leading to increased health risks to their hosts.

In nature, parasite species form diverse communities in which their members often compete for host resources during within-host coinfections as well as for available hosts during between-host transmission[1–3]. There is growing evidence that coinfections are common in plants[4] and animals[5,6], including humans[7] and that they can influence various aspects of host-parasite interactions and disease epidemiology. Coinfections can affect the amount of damage inflicted on the host, in many cases increasing it[8–11]. Yet, other studies demonstrated decreased damage[12,13] or no changes under coinfections[14]. These effects of parasite coinfections can also result in a modified transmission rate[11,15]. Additionally, transmission can be altered by the effect of coinfection on parasite reproduction[16–18]. Therefore, understanding the factors that determine the prevalence of coinfections is important for improving epidemiological modeling as well as disease management capabilities.

Since interactions among coinfecting parasites can generate facilitative[19] or inhibitive[20,21] effects on the participants, they may serve as an important factor in determining the frequency of coinfections. If interactions among the parasites are strong, the frequency of coinfections should be different from what is expected by chance[22]. In one example, the negative association among gut helminths and bovine tuberculosis in populations of African buffalo resulted in a lower frequency of coinfections than expected by a null model prediction[23]. However, the frequency of coinfections can

simply be a derivate of the prevalence of the different species in the parasite community. In such a case, heterogeneity in parasite prevalence should greatly influence the frequency of coinfections. The prevalence of parasites often varies over time[24,25], in space[26,27], or both[28–30]. Several studies demonstrated heterogeneity in coinfection occurrence or risk over time and in space[31–36]. Revealing whether interactions among parasites or simply their prevalence is the dominant driver behind coinfection frequency is a step towards understanding patterns of coinfection frequency in natural populations.

Here we studied the determinants of endoparasite coinfection frequency in natural populations of the water flea *Daphnia magna*. *D. magna* is a small crustacean that inhabits freshwater bodies in the northern hemisphere. It is naturally infected by a large variety of endoparasites[37–40] by ingesting their transmission stage (i.e., spores) from the environment while filter feeding. These parasites vary in the amount of damage (i.e., virulence) they inflict on their hosts, from having a small negative impact to causing a major reduction in reproduction, including castration, and longevity[37,41–43]. By infecting different sites within the *Daphnia* body, endoparasites can be classified into three categories: gut parasites, intracellular internal parasites (e.g., those infecting the gonads, fat cells and hypoderm) and extracellular internal parasites (e.g., those infecting the hemolymph and body cavity)[37]. Several studies have examined the interactions among endoparasites of *Daphnia* (Table 1).

[1]School of Zoology, George S. Wise Faculty of Life Sciences, Tel Aviv University, Tel Aviv 6997801, Israel. [2]Department of Evolutionary and Integrative Ecology, Leibniz Institute of Freshwater Ecology and Inland Fisheries, Berlin, Germany. [3]Department of Biology, Chemistry, Pharmacy, Institute of Biology, Freie Universität Berlin, Berlin, Germany. ✉e-mail: snirhalle@mail.tau.ac.il

**Table 1 | List of endoparasite species identified in this study**

| Species | Group | Target tissue | Category | Virulence | Changes under coinfections | References |
|---|---|---|---|---|---|---|
| Agglomerata sp. | Microsporidia | Hypoderm and fat cells | Internal parasites (intracellular) | Unknown | Unknown | 100,101 |
| *Hamiltosporidium tvaerminnensis | Microsporidia | Gonads and fat cells | Internal parasites (intracellular) | Horizontal transmission- no significant effects on the host. Vertical transmission- reduction in life span and fecundity. | Horizontal–cannot establish infection when facing P. ramosa. Vertical–reduced spore production. Damage to the host–greater reduction in fecundity when coinfected with P. ramosa; Greater reduction in longevity with P. ramosa. | 16,37,42,102,103 |
| Metschnikowia bicuspidata | Fungi | Body cavity and hemolymph | Internal parasites (extracellular) | Major reduction in life span and fecundity | Reduced spore production when it arrives after O. colligata. | 37,44,104 |
| Spirobacillus cienkowskii | Bacteria | Hemolymph | Internal parasites (extracellular) | Major reduction in lifespan | Unknown | 37,105,106 |
| Pasteuria ramosa | Bacteria | Hemolymph | Internal parasites (extracellular) | Reduction in life span and castration | Prevent infection of H. tvaerminnensis unless it was present vertically. Reduced spore production with H. tvaerminnensis or with M. bicuspidata. Damage to the host- greater reduction in longevity and fecundity with H. tvaerminnensis | 16,37,45,107 |
| Glugoides intestinalis | Microsporidia | Gut | Gut parasites | No significant effect on the host | Unknown | 37,41,108,109 |
| Ordospora colligata | Microsporidia | Gut | Gut parasites | Small reduction in the host's reproduction | Reduced spore production when arrives before M. bicuspidata | 37,41,44,110,111 |
| Unfamiliar gut microsporidium | Microsporidia | Gut | Gut parasites | Unknown | Unknown | NA |
| Pansporella perplexa | Amoeba | Gut | Gut parasites | Unknown | Unknown | 46,93 |
| Unknown microsporidium Spp. | Microsporidia | Unknown | Internal parasites | Unknown | Unknown | NA |

*While all parasites of D magna described in this study transmit horizontally, H. tvaerminnensis can also be transmitted vertically. Since transmission mode affects the interaction with the host, they are differentiated in the table where relevant. The site of infection, as well as, parasite virulence, and outcome of coinfections (as known from the literature) are presented.

Regardless of the site of infection, coinfection usually results in decreased reproductive output of both competitors. However, this outcome is influenced by the order of arrival to the host[16,21,44,45]. Additionally, within-host competition between differently virulent parasites can limit the ability of the less virulent parasite to successfully infect the host[16]. For example, the microsporidium *Hamiltosporidium tvaerminnensis* cannot infect its host when facing a more virulent parasite unless it is vertically transmitted to the host[16]. Host longevity under coinfection is usually determined by the more virulent competitor[16,44,45]. Additionally, in comparison with a single infection, coinfection can result in greater fecundity reduction for the host[16]. This reduction was also shown in natural populations[46]. In Israel, *D. magna* primarily inhabits ephemeral ponds that are characterized by seasonal proliferation of endoparasites and coinfected individuals are commonly observed[39]. Therefore, these populations offer a unique opportunity to study the role of the different drivers of coinfection frequency, given their potential for both variations in parasite prevalence and significant interactions among endoparasites.

Our aim was to assess the prevalence of coinfections in natural host populations and identify the main factors influencing their frequency. To this end, we combined descriptive and analytical approaches on the presence/absence data of endoparasites within their individual hosts. First, we performed a longitudinal field survey of eight populations of *Daphnia magna*, spanning from the appearance of host individuals in the ponds until their extinction. This enabled us to quantify the prevalence of coinfections within each population as well as the changes that occurred during the season. Due to the temporality of the habitat as well as temporal changes in host populations and parasite communities[39,47], we predicted that coinfections would become more frequent over time. Second, we tested whether the observed prevalence of coinfections differed from what is expected by chance, by using a null model approach[22]. We then applied a network analysis of the sampled parasite communities, to reveal potential nonrandom interactions among species and their nature[48]. Furthermore, we performed a model selection on a set of mixed-effect models in an attempt to identify the factors influencing the likelihood of coinfections. This combined approach allowed us to identify the role of parasite interactions versus stochasticity in determining the frequency of coinfections. Here we have two separate predictions. If parasite interactions strongly affect coinfection frequency, we predict a lower prevalence of coinfections than what is expected by chance. This is due to the negative effects of *Daphnia* parasites on each other, as observed in the lab[16,45]. However, if parasite interactions are not dominant, we predict that coinfection prevalence will be similar to the expectations of the null model and will generally reflect the prevalence dynamics of the parasite community members. A secondary objective was to understand the implications of coinfections on *D. magna* populations. We used linear mixed-effect models to test for the effect of parasitism in general and coinfections in particular on host density. Since many parasites of *D. magna* are known to have a negative effect on their host populations[41,47] we predict that parasites will have a negative effect on the populations in our ponds as well. However, laboratory studies demonstrated that the survival of individual hosts is not affected by coinfections[16,44]. Therefore, we predict that the negative effect of coinfections will be similar to the effect of infections by a single species.

## Results

Our total sampling effort, in eight *D. magna* populations (Table 2), resulted in 4440 *D. magna* individuals being dissected and nine species of microparasites (Table 1) being identified (one of which is probably new to science). In addition, we pooled unidentified microsporidia encountered in low frequencies and treated them as a single column ("unknown microsporidia") in the dataset. To avoid bias from samples with low sample sizes, we excluded from the analysis samples with fewer than 20 *D. magna* individuals. As one of the populations was left with only two samples, we excluded this population as well, resulting in 4324 *D. magna* individuals from seven populations in the final dataset (Table 2).

### Characteristics and frequencies of coinfections

We found coinfection in 33.28% of the infected *D. magna* individuals. The frequency of coinfections increased over time, becoming as (or more) prevalent as single infections in five out of seven populations (Fig. 1). This was in line with the dynamics of parasite communities that reached their highest prevalence and diversity in the later parts of the season (Fig. 2). In accordance, the prevalence of coinfections was highly correlated with the prevalence of parasitism in the population (Spearman's Rho = 0.82). However, this relationship was not entirely linear, as we never observed coinfections when parasite prevalence was below 40% (Fig. S1). Coinfections mainly comprised two parasite species (26.29%), followed by coinfections by three (6.25%), four (0.61%) and five (0.13%) parasite species. Furthermore, coinfections during winter tended to be comprised solely of gut parasites, while coinfections during spring were comprised mainly of both gut and inner body parasites. Coinfections comprised solely of inner body parasites were rare (Fig. S2).

### Null model, network analysis, and model selection

The prevalence of coinfections in the populations was highly correlated with the predictions of the null model ($r = 0.995$; Fig. S3; a preliminary test of a model without limitation on the maximal number of infecting parasites yielded similar results), i.e., the observed prevalence of coinfection was not different from what is expected by chance. Moreover, this correlation remained high throughout the randomization process of the data (mean ± SE $r = 0.995 ± 0.00003$; see Supplementary Data). In accordance with these results, exploration of parasites' interaction networks (Figs. 3 & S4–S9) within each host population for each sample revealed that in most cases, parasites with the highest prevalence ended up forming the edge with the highest weight. Moreover, the sum of weights from all the edges a parasite was involved with was in strong correlation with its respective prevalence (Pearson's $r = 0.66$; Fig. 4). Repeating this analysis separately per species revealed an even stronger correlation (mean ± SE Pearson's $r = 0.77 ± 0.07$; Fig. S10). However, prevalence was less related to the ability of a parasite to have diverse connections (i.e., the number of other species it

## Table 2 | Sampling sites and their respective number of samples in which *Daphnia* was detected

| Pond | Coordinates | First sample (date) | Last sample (date) | Number of samples |
|---|---|---|---|---|
| Poleg swamp 1 | 32°15'20.5"N 34°51'11.6"E | 17/02/19 | 17/03/19 | 3 (0) |
| Poleg swamp 2 | 32°15'25.5"N 34°51'15.3"E | 17/02/19 | 06/05/19 | 7 (6) |
| Poleg swamp 3 | 32°15'15.0"N 34°51'16.1"E | 17/02/19 | 14/04/19 | 4 (4) |
| Soreq swamp | 1°56'12.6"N 34°44'21.5"E | 17/02/19 | 26/05/19 | 10 (10) |
| Ga'ash Kibbutz | 32°13'58.5"N 34°49'27.0"E | 18/02/20 | 18/05/20 | 9 (9) |
| Herzeliya park | 32°10'15.5"N 34°49'27.2"E | 18/02/20 | 11/05/20 | 8 (6) |
| Ya'ar pool 1 | 32°24'44.45"N 34°53'58.93"E | 18/02/20 | 11/05/20 | 8 (6) |
| Ya'ar pool 2 | 32°24'49.88"N 34°53'53.06"E | 18/02/20 | 19/04/20 | 4 (4) |

In parenthesis, the number of samples included in the analyzed dataset (samples containing <20 D. magna individuals were excluded; in addition, only two samples remained for "Poleg swamps 1", resulting in the exclusion of this site).

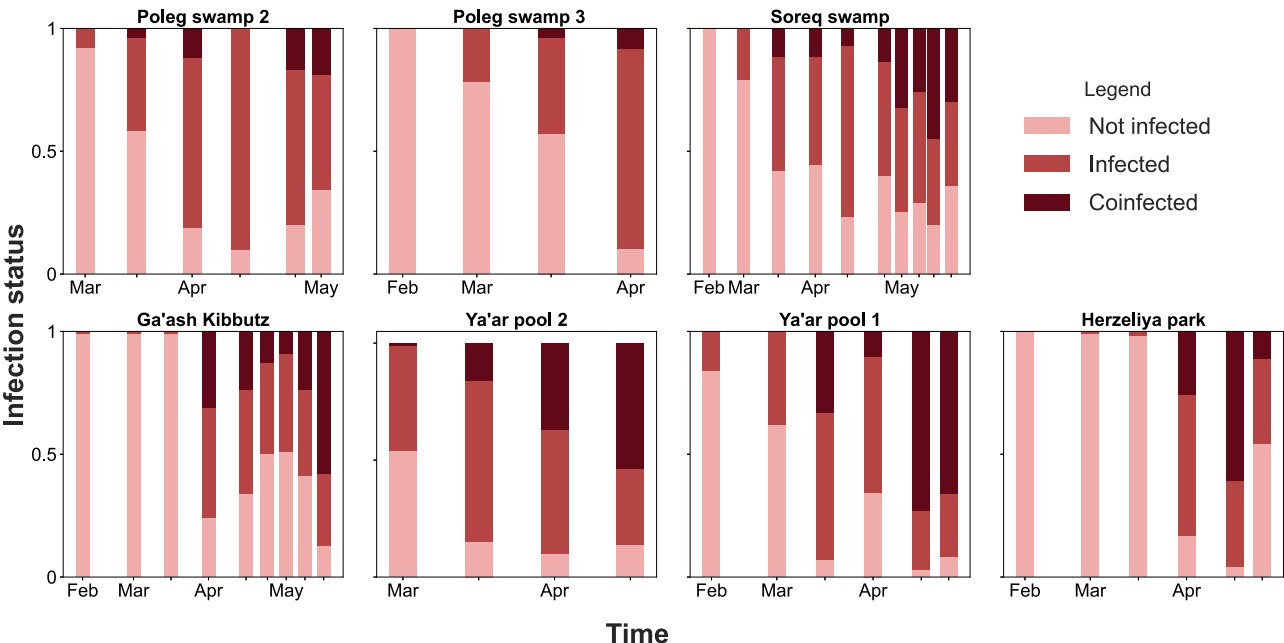

**Fig. 1 | Proportion of *D. magna* individuals with different infection statuses in the sampled populations over time.** The infection status is represented by a color gradient: not infected in pink, infected (i.e., infection by a single parasite species) in red, and coinfected (i.e., infection by at least two parasite species) in dark red.

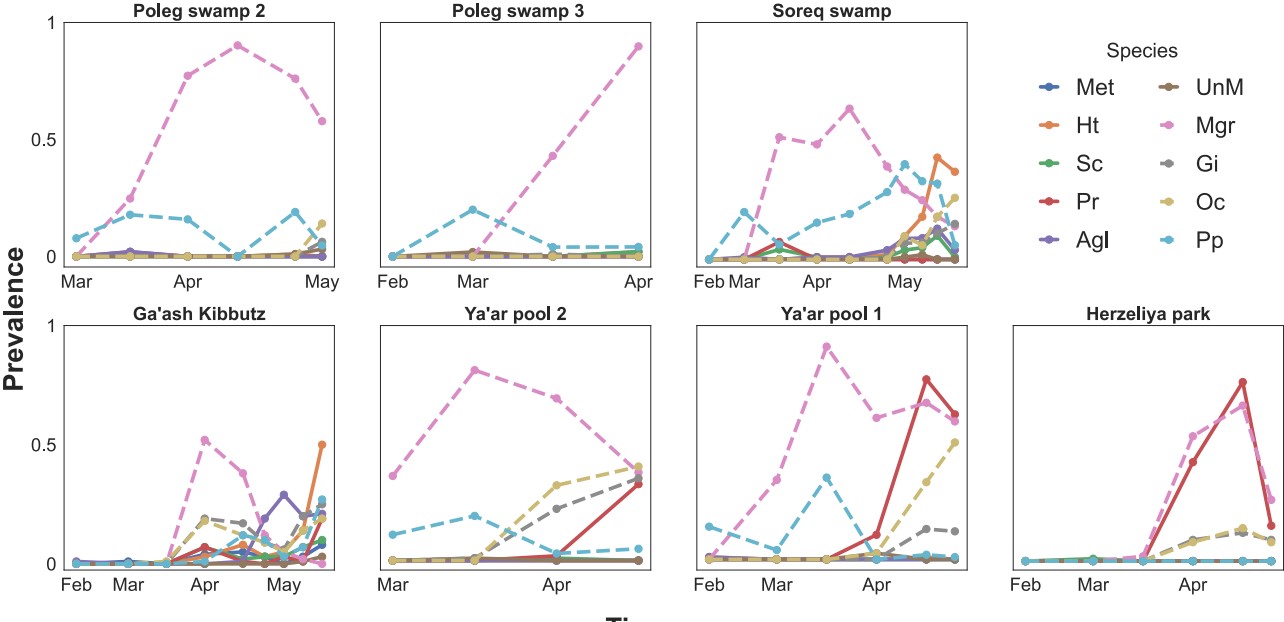

**Fig. 2 | Parasite species prevalence over time.** Dots on the lines represent the sampling date, and line type, regular or dashed, represents inner body or gut parasites, respectively. Species abbreviations are as follows: Agl- *Agglomerata* sp., Gi- *Glugoides intestinalis*, Ht- *Hamiltosporidium tvaerminnensis*, Met- *Metschnikowia bicuspidata*, Mgr- Unfamiliar species of microsporidium infecting the host guts, Oc- *Ordospora colligata*, Pp- *Pansporella perplexa*, Pr- *Pasteuria ramosa*, Sc- *Spirobacillus cienkowskii*, UnM- represents the pooled category of unknown microsporidia species.

was found to coinfect with). Parasites could achieve a high diversity of connections even at relatively low prevalence. The centrality of most parasites during the entire season was high, which indicates that even rare parasites were able to coinfect with most of their community members (see Supplementary Data). For most parasite species, there was a tendency for normalized centrality to increase with prevalence (Fig. S11). However, the overall strength of this positive correlation was relatively low (Pearson's $r = 0.41$; Fig. 4), thereby demonstrating that high diversity of connections could be achieved even if the prevalence of the parasite was low.

The results of the best model (Table 3) based on model selection (see Supplementary Data) highlighted four parameters that affect the coinfection rate: Simpson's diversity index, water temperature, host density and oxygen concentration. However, the predictors' estimate, as well as their Pearson's $r$ (a proxy of their effect size[49]) revealed that they are not equally important. While the estimates for Simpson's diversity index and water temperature were high and had high effect sizes ($r = 0.72$ and $r = 0.56$, respectively), the estimates and effect sizes of host density and oxygen concentration were low ($r = -0.33$ and $r = 0.21$, respectively).

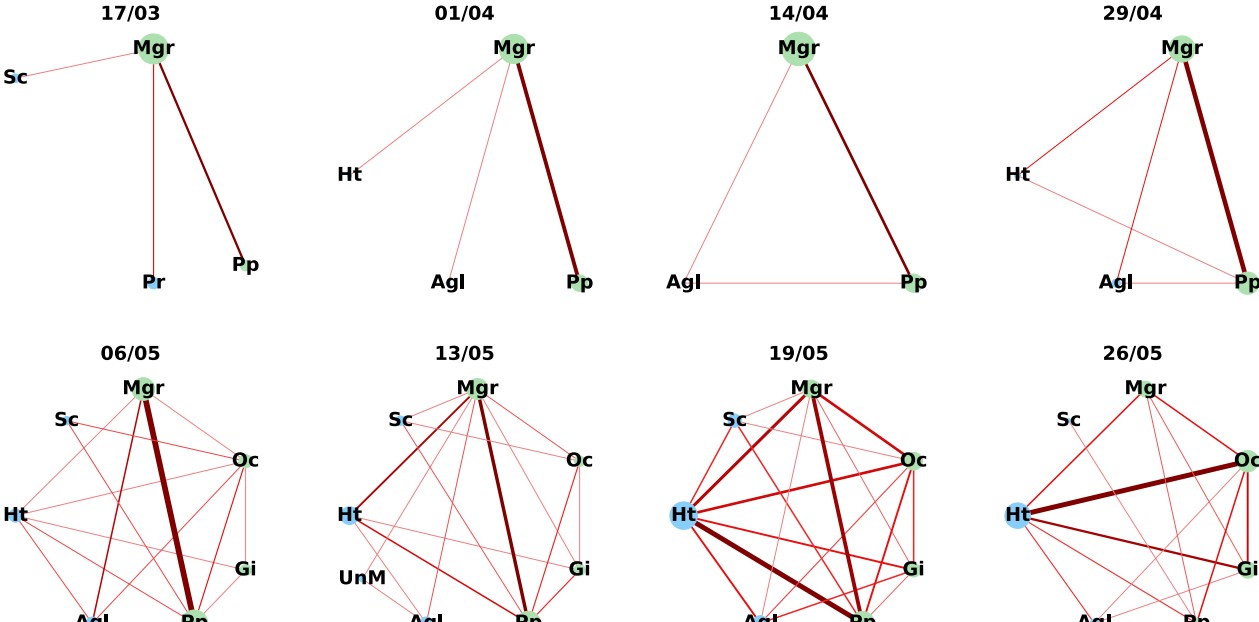

**Fig. 3 | Parasite infection networks in Soreq swamp, one of the sampled populations, throughout the entire season.** The date of the respective sample is presented at the top of each network. Nodes, i.e., parasite species, are represented by circles with species abbreviations. The prevalence of the parasite species is represented by the relative size of the circle. Edges, i.e., coinfecting pairs, are ranked relatively by color gradient and thickness of the line (edge weight), with lines becoming darker and thicker as a pair becomes more frequent. Parasitic groups are designated by different node color, green or blue for gut or inner body parasites, respectively. Species abbreviations are as follows: Agl- *Agglomerata* sp., Gi- *Glugoides intestinalis*, Ht- *Hamiltosporidium tvaerminnensis*, Mgr- Unfamiliar species of microsporidia infecting the host guts, Oc- *Ordospora colligata*, Pp- *Pansporella perplexa*, Pr- *Pasteuria ramosa*, Sc- *Spirobacillus cienkowskii*, UnM- represents the pooled category of unknown microsporidia species. Infection networks for the other host populations are presented in Supplementary Figs. (S4–S9).

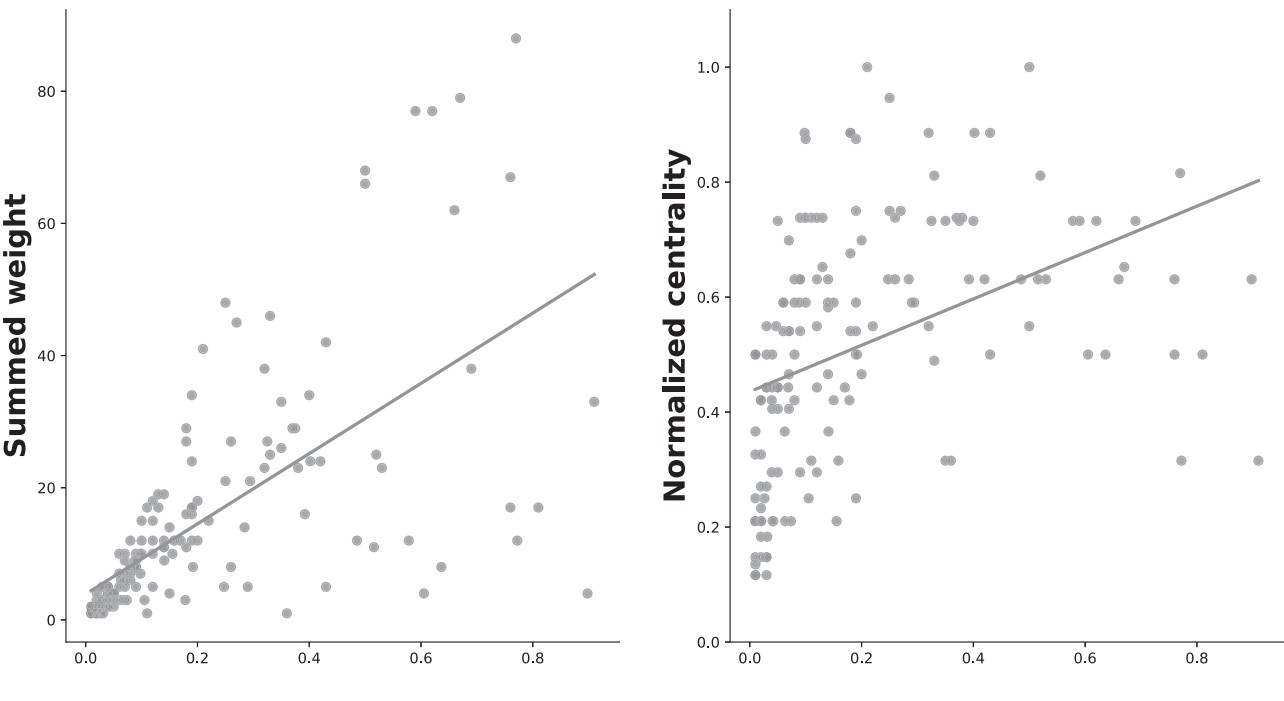

**Fig. 4 | Correlation between parasite species prevalence and infection network metrics.** Parasite species prevalence is correlated against its summed weight or normalized centrality (left and right hand, respectively) derived from the network analysis of every sample in every pond. If a parasite was absent from a sample (i.e., prevalence = 0), it was omitted from the analysis to avoid zero inflation of the data.

**Table 3 | Results of the best generalized linear mixed-effect model, testing for the importance of various predictors on the likelihood of coinfection, identified by model selection (AICc)**

| Parameter | Estimate | SE | Z value | Effect size |
|---|---|---|---|---|
| Intercept | −0.8852 | 0.2219 | −3.988 | NA |
| Host density | −0.2755 | 0.1255 | −2.196 | −0.33 |
| Simpson's diversity index | 1.0840 | 0.1649 | 6.574 | 0.72 |
| Water temperature | 1.3763 | 0.1601 | 8.596 | 0.56 |
| Oxygen concentration | 0.2211 | 0.1352 | 1.636 | 0.21 |

This model included the identity of the pond as a random intercept effect as well as a random slope effect for species richness. All model's values are presented after scaling the parameters for increased interpretability. Additionally, the effect size of the parameters (estimated by Pearson's *r*) is presented. The complete list of models and their information criteria can be found in the Supplementary Data.

## Host density

Our linear mixed-effect model testing (accompanied by Pearson's r as an estimate of the effect size) showed that parasite prevalence negatively affected the density of their host ($p = 0.001$, $r = −0.45$; Fig. S12). Coinfections were a significant contributor to this effect ($p = 0.024$, $r = −0.33$; Fig. S12), while single infections were found to be insignificant in the model despite having a similar effect size ($p = 0.065$, $r = −0.35$; Fig. S12). The gut parasites in our study were found to have a significant negative effect on their host density ($p < 0.001$, $r = −0.49$; Fig. S13), but the other parasite species had no effect ($p = 0.757$, $r = −0.17$; Fig. S13). Seasonal changes in host density in each pond, along with the prevalence of coinfections and infections by a single parasite species, are presented in Fig. S14. All the abiotic parameters measured in our study turned as insignificant in the preliminary testing (see Supplementary Data).

## Discussion

The commonness and importance of coinfections have been highly acknowledged in recent decades, underscoring the importance of our understanding of the determinants of coinfection risk in natural populations[7,50–52]. We studied the determinants of coinfection frequency in natural populations of *D. magna*. Specifically, we identified natural patterns of coinfection prevalence in these populations and investigated whether coinfection prevalence is merely a reflection of parasite prevalence or if it is influenced by parasite-parasite interactions. When the prevalence of parasites in the population was below 40%, no coinfections were present. As parasite prevalence increased with time, so did the prevalence of coinfections, resulting in a strong temporal pattern and increasing prevalence of coinfections towards the end of the season. Additionally, the prevalence of coinfections was not different from what is expected by chance, and the associations among parasite species were random. Taken together, coinfections in natural populations of *D. magna* seem to depend largely on the prevalence of parasite species.

While seasonal patterns in parasite prevalence are featured in many host-parasite systems[24,25], their relation to patterns of coinfection is often overlooked (but see refs. 32,36). Unless disrupted by strong negative interactions[11,53], parasite species with overlapping epidemic curves have a higher chance of becoming involved in coinfections. Under such scenario, the prevalence of coinfection is expected to be higher when the prevalence of several species peaks at a similar time[32,36]. In our study, most parasites increased in prevalence at the same time, resulting in many overlapping peaks of prevalence (Fig. 2). This led to an increase in coinfection prevalence that became the dominant form of infection in some host populations (Fig. 1). Moreover, our analysis showed that Simpson's diversity index was a strong predictor of coinfection (Table 3). This diversity index emphasizes prevalence rather than species richness per se[54] and further supports the

importance of overlapping in the prevalence of multiple parasite species as a driver of coinfection.

The relationship between overlapping parasite prevalence and coinfections should be relevant for other systems as well. For instance, coinfecting avian malaria and West Nile virus share overlapping prevalence patterns in American robins and their mosquito vectors[55]. Other vector-borne parasites also have the potential to demonstrate a link between prevalence patterns and coinfection, especially due to the temporal variation in vector abundance[56–59]. Additionally, the high turnover rate of the host can form strong seasonal patterns even for hosts with more complex immune responses or long-lasting parasites[60,61]. For example, the prevalence of wood mice parasites changes along the season, and though not tested, coinfections are unavoidable due to the overlapping prevalence of various parasite species[60]. Therefore, monitoring seasonal patterns of parasite diversity can be a useful means for estimating the risks of coinfections for a variety of natural host populations.

Temperature is an intrinsic characteristic of seasonality, and changes in temperature are linked to seasonal patterns of many organisms, including parasites (see examples in Table 1 of ref. 24). In our system, temperature plays a significant role as it affects both host development and physiology[62–65] as well as host-parasite interactions[66–70]. We found water temperature to be an important predictor of coinfection (Table 3). Elevated temperature leads to increased filtration rate in *D. magna*[65,71], which is expected to increase the intake of spores and, therefore, the likelihood of infection[72]. In addition, *Daphnia* parasites generally develop and infect better in warmer, but not too warm, temperatures[70,73,74] (but see ref. 75). Indeed, most of the parasite species in our populations increased in prevalence during the warmer spring season (Fig. 2). However, further research is needed to understand if temperature influences coinfections in ways beyond simply increasing the probability of infection.

Our populations developed in an unstable habitat (i.e., ephemeral pond) that is difficult to predict, and thus the parasites' window of opportunity is probably narrow. This is because the length of the ponds' hydroperiod is stochastic, and so is the persistence of host populations (see variation in the number of sampling events in Table 2). These circumstances might have set the stage for the seasonal patterns in parasites and coinfection prevalence observed in our study, as many parasites were "forced" to proliferate simultaneously rather than evolve temporal segregation. In a world subject to climate change that is expected to increase habitat instability[76–78], parasites may face shorter activity periods which might force them to overlap in time. This may lead to new risks of coinfection by parasites that used to be temporally segregated[79]. Future studies comparing the dynamics of parasite communities in stable vs. unstable habitats will increase our understanding of the relationship between habitat stability, multi-parasite dynamics and the risk of coinfections.

Null models have been used to estimate the risk and prevalence of coinfections in various host-parasite systems[80–82], yielding both random and non-random likelihoods of coinfections. Interestingly, even similar systems generated different results[81,83]. Since most of these studies analyzed species in pairs (even when studying the whole community[82]), it is possible that analyzing the parasite community as a whole would have resulted in random associations due to the potential of positive and negative associations to cancel each other out[84]. Here we accompanied our null model with parasite interaction networks in an attempt to unravel possible associations[48]. The analysis revealed a strong correlation between parasite prevalence and the structure of the network (Fig. 4), as the most common combinations of coinfection were frequently formed by the prevailing parasite species (Fig. 3 and S4–S9). Therefore, this analysis further supports that parasite prevalence generates the pattern of coinfections found in our system rather than parasite-parasite interactions.

In the field, mechanisms and processes other than parasite interactions (e.g., genetic interactions among hosts and parasites or environmental effects[85]) can reduce or even mask the effects of parasite-parasite interactions found in laboratory settings. Alternatively, they can alter the interactions among parasite species[86–88] and counter our lab-based expectations.

However, the neutral theory of community ecology suggests treating species within a community as equal and assuming that their recruitment is affected only by ecological drift and demographic stochasticity[89–91]. While not completely ignoring mechanisms that may underlie the resulting patterns, it claims that these patterns can be predicted without taking them into account[90]. The prediction of our null model demonstrated a high fit to the observed prevalence of coinfections (Fig. S3). Although more studies are necessary, there is much potential in applying concepts of the neutral theory on coinfection risk in parasite communities.

While parasite-parasite interactions among *D. magna* parasites have been observed in the lab[16,44], our null model, followed by network analysis, did not detect any sign of such interactions in our natural populations. Nonetheless, some interactions could have taken place without leaving a footprint on the prevalence of coinfection or of specific species. The most common result of coinfection in this system is a reduction in parasite spore production[16,44]. Although such an outcome can result in lower transmission potential from an infected individual, from a population perspective, the effect might be insignificant. This is because spores from hosts infected by a single parasite or from the pond's sediment can compensate for the loss of spores from coinfected hosts. Furthermore, when coinfections are the common scenario, variation due to the order of arrival of parasite species in sequential infections[16,44] and various timings of acquiring the second parasite species can mitigate the negative impact of coinfections on spore production. Another interaction observed in the lab is that parasites with a high virulence profile (see examples in Table 1) can competitively exclude or deny infection of a less virulent competitor[16]. In ponds that experienced the highest prevalence of the virulent bacterium *Pasteuria ramosa* we observed a lower diversity of internal parasite species (Fig. 2). However, since these populations were few and became extinct before May, when most internal parasites were expected to appear, we cannot determine if the lower diversity is the result of a parasite exclusion rather than host extinction. These two examples highlight the gap between conclusions based on lab observations on individual hosts and their implications at the host population level, especially in natural settings. While some within-host effects (on host or parasites) have a significant impact that can be observed at the population level[11], other outcomes might be less relevant in a more complex setting. Additionally, in systems in which parasites are acquired via the environment, the effects of parasite interaction may be hidden in the gap between infection potential (e.g., availability of spores in the water column and sediment) and infection prevalence. Future studies incorporating methods that can monitor off-host-parasite stages can help us to understand how the presence of different parasites in the community or the prevalence of coinfection affects the transmission potential and prevalence of competing parasite species.

In line with previous studies[41,47,92], infections had a negative effect on host population density. Our analysis further demonstrated that coinfections play a significant role in this effect. While it is possible that coinfection will affect host density via effects on host survival[8,9,11,13], this is less likely for *D. magna* since host survival does not change under coinfections[16,44]. However, Stirnadel and Ebert[46] found that the fecundity of coinfected *D. magna* was significantly lower than that of singly infected hosts. It is possible that host density in our populations is regulated through the increasing effect of parasites on host fecundity. This can explain why coinfection yielded a larger estimate in the model than single infection, despite having a similar effect size (−0.35 and −0.33 for single infection and coinfection, respectively). Surprisingly, gut parasites were the only group to have a negative effect on host density, despite being considered of low virulence[37]. This group was the most common group in coinfection throughout the season (Fig. S2) and, therefore, may have had a greater share in the overall negative effect of parasites. Additionally, under natural conditions, these parasites may bear more costs than can be observed in the lab.

In conclusion, coinfections in *D. magna* populations showed a clear temporal pattern that resulted in a narrow time frame in which coinfections were as common as single infections or even exceeded them. We link this pattern to the overlapping dynamics among members of the *Daphnia* parasite community and emphasize the importance of this characteristic of the parasite community that can shape the nature of coinfections in host populations. Similarly, the increased probability of coinfections when parasites overlap in time is due to an increased likelihood of encountering another parasite within the same host. We further found that the rate of coinfections was not different from what is expected by chance and that the associations among parasites were random. Without excluding the effects parasites have on each other, we believe that simple models can be beneficial for assessing general patterns of coinfections, as stochasticity or balancing effects can result in neutral outcomes. Future studies should look at the conditions that tend to lead to random or non-random patterns of coinfection, including the strength of interactions, host life history and environmental conditions. Since many parasites exhibit temporal patterns in their prevalence[24], we strongly advocate studies that will increase our understanding of their relation to the risk of coinfection, as these patterns are pivotal for our ability to predict and control the risk of coinfection.

## Materials and methods
### Study sites and field sampling
Sampling took place between February and June of 2019 and 2020. We chose eight ephemeral ponds (four per year) from the coastal plain of Israel, each with known populations of *D. magna*[39] (Table 2). We sampled the populations every other week during the winter, when diversity and prevalence were expected to be low, and every week during spring, when parasite diversity and prevalence are known to increase[39]. The first sampling event in each season took place when the *D. magna* populations had already developed in the ponds, but their parasites were still absent or at very low prevalence. We sampled each pond until we did not find *D. magna* individuals in two consecutive samples.

### Abiotic parameters
On each sampling event, we measured water temperature, dissolved oxygen concentration (mg/l), specific conductance (mS/cm) and pH using a YSI ProPlus multi-parameter instrument (Yellow Springs, Ohio, USA). We submerged the probe in the pond, in close proximity to where the live samples were taken, and recorded the measurements when the readings from the device had stabilized.

### Population density of *Daphnia magna*
On each sampling event, we submerged a plankton net (opening diameter: 27 cm, mesh size: 200 µm) into the water and dragged it for a distance of 15 m to maintain a consistent sampling effort. We transferred the content of the net to a glass jar containing ethanol (96%) to preserve it for further processing. In the lab, we stirred each sample and transferred its content to a petri dish pre-marked with eight equal subsections. We placed the dish under a dissecting microscope for identification and quantification of the density of *D. magna*. When the density of animals in the sample was low, we counted all individuals identified as *D. magna*. Otherwise, we transferred an eighth to one-quarter of the sample to a second petri dish and counted all individuals therein. Thereafter, we log-transformed the density of the population for further analysis.

### Parasite screening
Following the density sample, we took an additional live sample to the lab and screened for parasites within four days after the sampling event. We randomly chose 100 adult females (if the number of females within a sample was smaller than 100, then all of them were screened) and checked each individual for morphological signs of infection under a dissecting microscope (Leica M205C). Thereafter, we dissected the individual and screened for parasites under a phase contrast microscope (magnification ×400; Leica DM2500 & DM500). We identified parasites by their spore morphology or unique infection characteristics[37,39,46,93]. To increase the detectability of gut

parasites, we removed the guts of some individuals and screened them separately. We executed this procedure in particular for *Daphnia* that demonstrated clear signs of infection, because infection causes loss of clarity of the *Daphnia* body and guts[37]. The use of this procedure varied according to the number of such individuals in a sample, from 0 to 100% of the screened individuals (mean ± SE: 33 ± 0.03%). However, this variation had no effect on the detectability of gut parasites (Fig. S15). For each host individual, we recorded the presence–absence of each parasite species. Additionally, we calculated diversity indices (Simpson's diversity index, Shannon's diversity index, species richness and species evenness) for the parasite community of each sample.

## Data analysis

To describe the rate of coinfections in general and over time, we calculated for each sample the proportion of uninfected individuals, individuals infected by a single parasite species and individuals infected by more than one parasite species. We then evaluated the changes in these proportions within each population over time.

To test whether the prevalence of coinfections is different from what is expected by chance, we correlated the observed prevalence of coinfections against a null model[22]. We constructed the model by calculating the probability of infection by any possible combination of parasite species. We calculated these probabilities for each sample based on the observed prevalence of each species. Since we did not observe any case of infection by more than five species, the maximal combinations were limited to five species to improve the model's accuracy. Then, we summed the probabilities into categories (sum of binomials) based on the number of infecting parasites (from 0 to 5) and consolidated all categories of coinfections (from 2 to 5) into a single category. Finally, we used the Pearson correlation test to estimate the strength of the correlation between the observed prevalence and the model's predictions. To estimate the robustness of the results, we performed a restricted bootstrap on the data 1000 times. Each time, we randomly assigned the presence/absence of each parasite species for every individual host in the dataset. The only restriction was that the total prevalence of each parasite in every sample was set to its observed prevalence. We applied this restriction to keep the randomization process in touch with the natural behavior of the parasites, while allowing the infection status of the individuals (not infected/infected/coinfected) to be set randomly. We calculated the prevalence of coinfections in the randomized dataset as described above and correlated the prevalence against the model's predictions. We used the mean Pearson's r as a measure of the reproducibility of the results.

To estimate if the associations between parasite community members is non-random, we generated a set of parasite networks for each sample in which coinfections were detected. The nodes in each network represent the species present in the sample, while the edges represent if a pair of species was found in coinfection. To estimate the strength of the edges, we calculated their weights as the sum of hosts coinfected by each pair. We compared the identity of the maximal edge weight to the identity of the parasites with the highest prevalence. In addition, we calculated the centrality of each node to assess how parasites tend to be in coinfection with other parasite community members. To improve the assessment of the relation between parasite prevalence and their abundance in coinfections, we correlated the sum of edge weights connected to a given parasite in every sample against its respective prevalence. Similarly, we correlated nodes' centrality against the prevalence of parasites to assess if the diversity of connection (i.e., with how many community members a parasite interacted) changed in relation to prevalence. However, since the number of members available in the parasite community changed among populations and samples, we penalized the centrality of samples relative to species richness. We calculated this normalized version of centrality using the following formula $NC_{ij} = C_{ij} * \ln(\text{rich}_i)/\ln(\text{rich}_{max})$, where $NC_{ij}$ is the normalized centrality, $C_{ij}$ is the degree centrality (i.e., the proportion of connections within the network), $\text{rich}_i$ is parasite richness, $\text{rich}_{max}$ is the maximal richness observed in all the samples, and $i$ and $j$ are the sample and the node, respectively. This process allowed us to account not only for the relative diversity of connections, but also for the difficulty of achieving them. Additionally, we calculated the degree centrality of the parasites from networks of the whole season in each population to obtain a wider perspective of this characteristic.

We tested the effects of abiotic and biotic parameters on the probability of an individual becoming infected by more than one parasite species using a generalized linear mixed-effect model with a binomial distribution and model selection approach. We generated a subset of the dataset containing only infected individuals and used the infection status (single infection or coinfection) as the dependent variable. In general, we constructed three sets of models following three hypotheses: (a) only changes in abiotic parameters affect the probability of coinfection, (b) only changes in biotic parameters affect the probability of coinfection, and (c) both abiotic and biotic parameters play a role in this phenomenon. Since all abiotic parameters besides oxygen concentration were correlated with time, we used each of them in separate models along with oxygen concentration. In addition, due to the relatively low number of populations in the sample, we used each diversity index solely in each relevant model. For example, a model from the last group will contain oxygen concentration, host density, additional abiotic factor (water temperature, specific conductance or PH) and a diversity index (Shannon's, Simpson's, evenness or richness) as fixed factors. We used the identity of the pond as a random intercept in all models. The complete list of models is available in the supplementary material. We ranked the different models based on their AICc value and reran the best model with scale alignment of the relevant parameters for increased interpretability.

Lastly, we tested the effect of parasites on their host density using a set of linear mixed-effect models. Specifically, we asked if infection prevalence affects host density. Then, we tested for the effect of both single infections and coinfections. Finally, we tested for the effect of each group of parasites (gut or inner body). Once again, we used the identity of the pond as a random intercept in all the models. To ensure we do not miss any effects on host density, we also tested for the effects of the four abiotic parameters (water temperature, oxygen concentration, specific conductance and pH). Since most of the parameters measured in this study were correlated with time, including parasite prevalence (see results and Fig. S1), we tested for each of them separately.

Data analyses were carried out in Python 3.8 using the Pandas[94], NumPy[95], SciPy[96], and NetworkX[97] modules. These included the null model construction, bootstrap, Pearson test, and network analysis. We calculated generalized linear mixed-effect model, linear mixed effect model and model selection in R 4.0.5 using lme4[98] and MuMIn[99] packages, as well as the base packages. We constructed all the figures in Python 3.8 using the Matplotlib, Seaborn and NetworkX modules.

## Reporting summary

Further information on research design is available in the Nature Portfolio Reporting Summary linked to this article.

## Data availability

The datasets supporting this article are available on Figshare via https://doi.org/10.6084/m9.figshare.25407505.v1. The numerical sources for Figs. 1 and 2 are also available in the Supplementary Data.

## Code availability

Both Python and R scripts that were used for data analyses and figure generation of this work are available on Fighshare via https://doi.org/10.6084/m9.figshare.25407505.v1.

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

## Acknowledgements

We thank Ofir Levy and Jonathan Belmaker for their valuable advice on the statistical analysis. We acknowledge the valuable time and help during the sampling seasons by members of Frida Ben-Ami's lab. A special thanks is dedicated to Elena Halle Rogovin for her useful comments and edits on the manuscript as well as her support when the living room turned into a lab during the COVID-19 lockdown. We acknowledge funding from the German Science Foundation (DFG) through a joint German–Israeli project (0604317501 to F.B.A., WO 1587/8-1 to J.W.).

## Author contributions

F.B.A. and J.W. obtained funding for this study. S.H. and F.B.A. conceived and designed the experiment. S.H. and O.H. collected the samples. S.H. sorted and processed the samples. S.H. analyzed the data with input from F.B.A. S.H. wrote the first draft of the manuscript. F.B.A. helped draft the manuscript and provided editorial advice. J.W. and F.M. provided comments on the manuscript. All authors gave final approval for publication.

## Competing interests

The authors declare no competing interests.

## Ethics

This study adheres to local guidelines. No ethical approval was required to conduct it. All samplings were conducted under permits 2019/42360 and 2020/42579 from the Israel Nature and National Parks Protection Authority.
