## [Peer Review File · Communications Biology]

Reviewers' comments:

Reviewer #1 (Remarks to the Author):

Summary: This study presents a temporal prevalence and coinfection data set collected from eight natural *Daphnia magna* populations and their parasite communities. The authors ask the interesting question of whether coinfections are mainly formed due to the prevalence's of the parasites or whether parasite-parasite interactions and environmental variables drive the coinfection patterns. They use three modelling approaches (null model, network analyses, GLMM) to test these questions and find that 1. coinfections follow those expected based on prevalence, 2. prevalent parasites form coinfections more often -although their co-occurrences are not necessarily more diverse- than parasites with low prevalence, and 3. among the infected hosts, coinfections were positively associated with Simpsons diversity and sampling time, and negatively with host density.

General comment: I read this manuscript with interest and was excited to see the results. I especially liked the figures of the development of the networks over time which I do not remember seeing previously, at least not often. The populations have been sampled through time to gain a better view of the development of the prevalence instead of relying on snap-shot data. The spatially and temporally limited host populations seem optimal for testing the questions because of the clear epidemic season. The data collection methods and analysis seem sound, the introduction was clear and nice to read. The methods and results were also mostly clearly written. A few concerns did arise that I would like to see clarified, especially some inconsistencies in the Methods, and a justification for the 4 species maximum set in the null model. Also, a more elaborate description of the system would be welcome to help with the assessment and interpretation of the results. In the Discussion, the focus and flow could be improved in some areas.

Detailed comments

1. There is currently inconsistency in description of the GLMM models. Variables listed in the main text and variables listed in the supplementary table do not match. Supplementary table lists also evenness and richness variables where the main text indicates that only Shannon diversity was used as fixed effect. Hence, it is unclear what models were fitted. The AICc values are also not presented (I expected to find them in the supplementary table of the model variants) despite being an integral part of the analyses and results.

2. The decision to limit coinfections to 4 parasite species in the null model is not currently explained. I am hence hesitant about whether the null model truly represents a random model. What if the observed maximum of 4 parasites in the data is lower than the null model would suggest if there was no limitation? Wouldn't there then be less coinfections than expected by chance and hence parasite interactions would be interpreted as influential?

3. A reader who is not familiar with the *Daphnia* parasite community would greatly benefit of a description of the system in the methods. For example: which parasites are present or expected to be present in the system, what is known about their interactions from lab studies, and how dynamic their presence in the populations are among seasons. These are hinted at (Line 117-118) "the parasites of *Daphnia* have negative impact on each other", but are these the same parasites as in this study? It seems they may not be because one of the parasites is described as "potentially new to science" (Line 247) and Discussion refers to "the low virulence of the parasites" several times. With more information, it would be easier to take in the Results and Discussion.

4. Discussion would benefit from some refocusing and polishing. Currently, the focus seems a little bit of here and there. The paragraph discussing host density in the GLMM model (Line 361-379) seems

contradicting to what the model tested for. The paragraph discusses how host density is affected by parasites. However, in the GLMM model host density is used as an explanatory variable to explain parasite coinfections. Hence, the paragraph seems out of scope in the middle of the paragraph while the ending provides some more expected discussion.

Another paragraph that the authors could reconsider is the hypothesizing about virulence evolution (Line 437-Line 454). While I agree that evolutionary effects are important and worthy of mentioning in the manuscript, in its current form and in this specific manuscript, this section seems a bit excessive and out of scope as the study does not specifically address these questions. The paragraph does not seem to discuss a result of this study. In addition, due to the short space provided for the topic, there is no space to discuss the full extent of the questions. Given that the Discussion is already on the longer side, this paragraph could be shortened into one or two sentences and combined to a conclusion paragraph. Or, the point that temporal variation can have evolutionary consequences, could be moved to the section in the Discussion that goes through the temporal patterns in the results. On the other hand, what I felt was missing is some discussion about what previous studies investigating parasite vs. coinfection prevalence. Is testing this question a neglected topic compared to other mechanisms (host-parasite, and environment drivers) or a frequently addressed topic? Null models have been around for some time (manuscript cites Gotelli 2000) and have also been used to study parasite metacommunity structures also previously (I think at least by Robert Poulin, and Joshua Brian and David Aldridge). Some summary about this would be welcome to help the readers not familiar with the null modeling literature. Similarly, examples about parasite epidemics peaks are surprisingly few in number. If this is because they are rare, it would be worth it to highlight it.

Line 29-30 Seems to be missing a word, perhaps "of"

Line 156 How often was this the case? I did not find these data. Is it presented somewhere or could it be added? For example into Table 1

Line 223 and 244 "For" -> "of"

Line 225 "abiotic variables listed above" It would be helpful to remind the readers what the variables are. If this means the variables listed under "Abiotic parameters", it does not correspond with supplementary table describing the models where it lists "w. temp", "time", and "SPC". There are also more variables in the supplementary table in the "biotic" models than described in the main text and it is hence difficult to conclude what models were fitted

Line 255 Would it be possible to provide the exact percentage instead of "about 20%"

Line 261: relation-> relationship

Line 237 I suggest adding the interpretation of this correlation here, which I suppose it is that the parasite interactions were random. Otherwise readers not familiar with the terminology used in null modelling will be confused to see this term used in the Discussion. It would be good to repeat it in the Discussion when the term first is used

Line 285 "The results of the model selection Table 2" I was surprised to not find the AICc comparison results in Table 2 (and later in the supplementary) due to the formulation of this sentence. "The results of the best model based on model selection" or something similar would be more appropriate here

Line 307 "end of the host activity season" A simple "the end of the season" would be sufficient and more in line with the terms used previously

Line 307 "increasing dominance" If I interpret the results correctly, coinfections only dominate infections in three populations in the end. Perhaps replace with "prevalence". Also the same situation on Line 257.

Line 308-309 Would be helpful to recite which modelling yielded these results

Line 313 Unnecessary words "to some extent"

Line 313-317 Please cite some examples

Line 322 Unnecessary word "dramatic"

Line 327 There seems to be a word missing or some other typo "that its"

Line 362 Unnecessary word "very"

Line 374 Suggest changing word "anyhow" to more formal "however"

Line 375 "low-virulent parasites" This would be relevant information to put with a more detailed descriptions about the parasites

Line 383-384 "...was illustrated by an almost perfect match between the identity of the most common coinfection combos and the most prevailing parasite species" This sentence is a little complicated and I had to read it multiple times to understand it. Would there be a more simple way to say this? Also "combinations" instead of "combos" would be more formal

Line 395-400 I am a little bit confused as to how the last sentences in this paragraph relate to the beginning of the paragraph. It almost seems as if they have been accidentally relocated to a wrong place in the Discussion. Would it be possible to improve the link and flow somehow

Line 410 "probably" is unnecessary here

Line 412 The neutral theory is introduced rather suddenly. If I understand the authors intention correctly, a "however" would be a fitting link word to contrast the idea that all mechanisms need to be taken into account to the idea of neutral theory? Adding a linking word would improve the flow

Line 417: "For example...r value of .." This result does not communicate much to a reader who has not read the original paper

Line 428 "Competitive ability is often determined by virulence" Some references of empirical examples would be welcome here

Line 461 "parasites overlap in space is due..." Do you mean time? I would assume time because it was part of the sampling design

Figure 3. Could you mention "population" in the first sentence or "Soreq swamp, one of the sampled populations". Because there are 8 network diagrams, it can first seem to be networks of each of the sampled populations instead of temporal points of one, and I got confused at first. It would also be helpful, if the node of the dominant parasite be at fixed position throughout the figures to make it easier to inspect the development of the network. Even better if the parasites in all figures would have

fixed positions to enable comparing among populations.

Figure 4. Please add to the figure text what analyses these correlations come from

Figure S1, S2, "Relation" should be "relationship"

Reviewer #2 (Remarks to the Author):

Summary: In this manuscript, the authors analyze infection prevalence of each of several parasites infecting ephemeral *Daphnia* populations. They find that rates of coinfection are consistent with a "neutral" model of parasite assembly within hosts, where the likelihood of coinfection of a given set of parasites simply increases with prevalences of the individual parasites in the population. Associations among parasite species were found to be random. The study provides important insights into patterns of coinfection through time in natural populations.

The manuscript is well written overall. I have several comments aimed at increasing the clarity further. My comments are listed in order from start to end of the manuscript. Some particularly important comments are preceded with two asterisks (**).

L1: I found the start to the title ("May the odds be with you") to be distracting, as it reminds me of famous quotes both from The Hunger Games books ("May the odds be ever in your favor") and from Star Wars ("May the force be with you"). I'm not sure if those references are intended. In any case, I think the title would be better simply as, "Coinfection frequency in water flea populations is a mere reflection of parasite diversity".

L30: Missing "of" in "natural dynamics of coinfection prevalence"

L35: Change "bound" to "bounds"

L47-48: None of the three papers cited here show that coinfections are common in plants. Please add another citation accordingly.

L52: do these "counter examples" show no effect of coinfection on damage to host, or a decreasing effect of coinfection on damage to host? I think the sentence starting at end of L49 should be ended after the citations in L51 and a new sentence started in L52 about the counter examples.

L55-58: I found this sentence to be confusing and possibly internally redundant. Could you simplify the start to: "Parasite competition can modify transmission, either directly by altering susceptibility or survival of hosts..."? Actually, the "survival" part of that makes sense to me with regards to modifying transmission, but I'm not sure how "susceptibility" is involved here if the parasites have already co-infected the host. Are you referring to susceptibility to another parasite that hasn't yet had the chance to infect? In the next part of the sentence ("...or indirectly by reducing host availability via a virulent community member...") I cannot tell how "reducing host availability" is different from "altering susceptibility or survival" in the first part of the sentence.

**L61: Please insert a description of the site of infection within host's body for each parasite. See also my comments below about somehow making this apparent in Fig. 2, S12, and S13. I had to dig through the supplemental data file to find that information in tab called "Parasite species list" (also there is a small typo in header of column D of that data sheet: "witihn").

L166-167: I'm confused by what Figure S1 is showing. I want to think of this dissection vs. prevalence result in terms of being able to detect rates of "false negative" and "false positive" assignments to infection status. But maybe that's not quite right because you don't know "true" infection status for individuals that weren't dissected? Clarification is needed on what we can learn from Fig. S1 in terms of how useful the gut dissections are for detecting gut parasites that wouldn't have been detected if you hadn't performed the dissection.

Something that would help with interpretation of Fig. S1: Are there multiple points overlapping at for example (0, 0), (0, 1), and (1, 1) positions on the plot? If so, you could scale the size of dots relative to number of overlapping points (e.g., using `geom_count` in `ggplot2` package in R). Also, I suggest changing x-axis titles of Fig. S1 and Fig. S2 to "Proportion of sample dissected" and y-axis titles to "Detected prevalence of gut infection".

L203-205: Suggested rewording for clarity: "To improve the assessment of the relation between parasite prevalence and abundance in coinfections, the sum of edge weights connected to a given parasite in every sample was correlated against its respective prevalence."

L212: The term "degree centrality" is defined in the caption of supplemental Fig. S13. Please also insert that definition here in main text.

L226: I assume "time in which the sample was taken" refers to date of year? But could also refer to time of day. Please clarify.

**L227-229: This is unclear in the "List of GLMM" tab of the supplemental data file. In that data sheet, the abiotic factors listed are "w.temp" (is that an abbreviation for water temperature? Or does the "w" stand for something else?), "SPC" (I know that is specific conductance, but that acronym needs to be spelled out in the table), and "time". Oxygen concentration is not listed as a factor in any of the models in that "List of GLMM" tab. But L229 of main text implies that oxygen concentration was included in some or all models with abiotic factors (unclear).

L256: Suggest deleting "Nonetheless" and simply starting sentence with "The frequency of...".

L257: Suggest replacing "dominant as" with "common than"

L266: Suggest inserting words "while" and "were" here: "...comprised solely of gut parasite, while coinfections during spring were comprised..."

L284-286 and Fig. S13: I found this wording to be unclear. Suggest changing from "For a given sample, there was a tendency of normalized centrality to increase with prevalence per species" to "For most parasite species, there was a tendency for normalized centrality to increase with its prevalence in the sample" if that is what you mean.

L286: Suggest rewording the description of Fig. 4 results to: "However, the overall strength of the positive correlation between prevalence and normalized centrality across all parasite species and samples was relatively low (Pearson's $r=0.41$; Figure 4)..."

**Discussion: The discussion section is very long. Currently nearly 7 full pages in total. I think two entire paragraphs could potentially be cut altogether, or at least greatly trimmed down: L401-421 (maybe you could weave in the nice references to neutral theory of community ecology elsewhere in Discussion?) and L437-454 (while interesting, the discussion of evolution of parasite virulence seems beyond the scope of this study).

L312: Delete "a" before "strong"

L316: Change "epidemic" to "epidemics"

L319: This is the only place where the term "intensity" is used, so please define here (or replace with "prevalence" if that is what is meant).

L324: Suggest starting a new sentence after "... (Figure 1)". For example, "In other populations, coinfection never became the dominant form of infection, possibly due to early extinction of those host populations before the vast proliferation of parasites."

L327: Suggest rephrasing to: "...was also the only species with a weak correlation between its prevalence and frequency of coinfections (Figure S12)."

L331-333: Suggest rewording to: "The Simpson's diversity index is an inverse of a dominance index. As such, Simpson's diversity index emphasizes the importance of overlapping prevalence of parasite species rather than their richness per se (Kim et al. 2017). In our study Simpson's diversity index was strongly correlated with time, ... " I the last part of L333 about how this "can explain diversity's high effect size on coinfections" also needs more explanation. Are you trying to point out that from your data set you can't tease apart the effect of Simpson's diversity index from effects of other time-correlated factors?

L344: After "additional parameters", consider adding "or additional geographic regions" to link with earlier in paragraph where you discuss limitation of sampling in single region of Israel.

L347-350: This sentence should be refined for clarity.

L384: replace "combos" with "combinations"

L391-393: This wasn't highlighted in results figures. But may be apparent if you indicate in figures which of the parasites infect hemolymph.

**L437-451: Above, I've suggested cutting this whole paragraph because it goes beyond the scope of the present study and the Discussion section is already quite long. (However, if I wasn't recommending cutting it, I would have commented that many of the ideas in this paragraph would require further explanation and citations. The logic was difficult to follow, particularly for L445-451 where there are no citations provided.)

Table 2: Consider replacing "Sampling Time" with "Sampling date". Also, suggest spelling out "Results of the best generalized linear mixed model (GLMM)..." in caption

**The results of the model competition aren't actually shown in the main text or supplement. Please add the key information (e.g., number of parameters estimated, AICc, delta AICc, Akaike weight, adjusted R²) into the "List of GLMM" tab of the supplemental data file and reference those model competition results in the main text.

**Figs. 2, S12, and S13: In the legend, could you please help us keep track of which parasite occupy which site in the host body? This could be done by grouping the parasite names by host site (instead of arranging them alphabetically) with additional labels in legend, or possibly with a shape aesthetic for site in host body.

Figs. S1-S2: Could you label the points by species of gut parasite, using same colour scheme as in

Figs. 2, S12, and S13? Can keep the one regression line through all species combined, but it would be nice to see if detection vs. dissection varies among gut parasite species.

Figs. 2 and S4: Please add small points to show the actual sampling dates at each site. Or if points are too hard to see amidst all the lines, you could add small tick marks on x-axes of all of these panels to indicate the sampling dates at each site.

Fig. S4: The colours are not appearing as described in caption. The "blue" appears black on my screen (maybe it's a very dark blue?) and the "teal" appears as more of a light blue.

Figs. S6-S11: I assume the networks are only drawn for dates where there were at least some (greater than zero) coinfections? This wasn't explicitly stated in the methods.

Figs. S12 and S13: "Prevalence" is misspelled in x-axis titles of both these figures.

Reviewer #3 (Remarks to the Author):

This paper uses a *Daphnia* model to investigate the potential for co-infections to be driven primarily by parasite prevalence rather than other factors such as interactions between co-infecting parasites. They found that the null model best predicts the frequency of co-infections and that overlapping parasite prevalence in the field directly predicts the likelihood that *Daphnia* would be co-infected. They also found that parasite density and time played a significant role in co-infection prevalence.

I find the investigation and concept of co-infection prevalence as simply a side effect of parasite prevalence interesting and I think they have strong evidence for this concept in their study system. However, their results are partially complicated by the significant effect of time in their model. Time plays a major role in this system, as *Daphnia* populations, parasite populations, and tested abiotic factors were correlated with time. While this is logical in this seasonal system, it makes it somewhat challenging to separate out if the results are driven exclusively by prevalence or also by other abiotic or biotic factors that were not tested. This is something the authors acknowledge. The ephemeral nature of this system makes me question how applicable these results would be to other systems, in which more complex co-infecting interactions may be more likely to develop due to the longer overlap of these parasites in the host system and/or through potential interactions with more complex host immune systems.

They find no effect of interactions between co-infecting parasites in their system. However, they compare co-infections in different parts of the host, which are unlikely to interact at all. Additionally, they acknowledge that the co-infecting parasite interactions that they do not find in their analyses could still be happening through exclusion from the system entirely. I would appreciate a more explicit explanation throughout the manuscript of which parasites or parasite types (gut vs. hemolymph etc) they anticipate interacting and what evidence they found to support that they did not. This is mentioned occasionally throughout the manuscript, but I believe a more thoughtful and thorough approach to this topic could strength the manuscript.

Finally, the paper starts out discussing the importance of co-infections on host health, parasite transmission, etc., but does not address those questions explicitly in the methods or results. There is some evidence from their methods that they could explicitly address these questions if they wanted (see detailed thoughts below). Their most prevalent co-infections included non-virulent parasites, which may mean that, although prevalence is driving these co-infections, these co-infections could play little to no role in individual host health or host populations. Virulence is also frequently brought

up in the discussion as being driven by co-infections or driving them, but which of the tested parasites were virulent and how this might be driving co-infections could be set up in the introduction and connected more thoughtfully in the discussion.

Line by line

Abstract:

Lines 30-32: This is quite a long sentence. You could add to the flow by breaking it up. I recommend "... or simply derive from parasite diversity. Identifying the relative impact..."

Line 36-37: I find this sentence a bit confusing. I think maybe clarifying the temporal pattern of the co-infection prevalence would help. Something like: "Additionally, coinfection prevalence increased over time, becoming equally common as single infections."

Lines 39-40: This statement overreaches the conclusions of your study as you did not address health risks. I think you could say something like: "We suggest that monitoring parasite diversity can help predict where and when coinfection prevalence will be high, potentially leading to increased health risks or increased parasite transmission."

Introduction:

Line 47-48: I would remove 'including humans'. Co-infections in this system differ in many ways from co-infections in humans.

Line 63: For clarity, please change 'If their effect' to 'If the interaction between the parasites'.

Line 72: I am unsure what is meant by "In return". I would remove this phrase.

Lines 90-92: This means that your co-infection system probably generates different dynamics than longer term systems in which parasites may be forced to interact with one another for growth/reproduction over longer periods of time. I would not list this among the reasons that this system is ideal for studying co-infections.

Lines 94-95: I am curious as to which endoparasitic species reduced each other's fitness. Can you specify the species tested? Also, a basic description of the common parasites or parasite types in daphnia and how you expect them to interact is needed. I think it will set up the hypotheses and results.

Methods:

Line 128: Add space before citation.

Line 157: Here you say that you check for 'morphological signs of infection', but there are no analyses on host health. Could these analyses be added? Understanding how co-infections or single infections altered host health in this study would help bolster the argument made in the first paragraph of the introduction. Co-infections matter when they alter host health or parasite transmission in some way compared to single infections.

Lines 225-226: Was parasite prevalence not included in this?

Lines 227-229: It would add clarity to specify which abiotic parameters were correlated with time here. By my understanding only four abiotic parameters were measured, so: "Since pH, water temperature, and conductivity were correlated with time, only one of these abiotic parameters or time was included in a single model along with oxygen concentration."

Line 234: The listed models in the supplement do not line up with the description offered in the manuscript. It is stated that only one abiotic variable (SPC, pH, or water temp) or time was included in the model. However, only models with SPC, water temp, or time were included. If those abiotic variables were too correlated to be included in a model together, why run 2 out of the 3 of them separately? Conversely, why not run all three? Additionally, despite explicitly stating on line 229 that oxygen concentration was included in the models, I do not see it included anywhere. Also, I would like some justification included in the manuscript for why specific variables that were correlated were not run in the same model (i.e., no abiotic factors except oxygen and time; line 227), but other correlated factors were (i.e., parasite density and time; line 335). If time is correlated with most other factors and there are no specific hypotheses about time, why is time included in the model?

Results:

Lines 266-268: This is an interesting temporal result that I do not believe is touched on in this discussion.

Discussion:

Many of the paragraphs in the discussion could be shortened or combined to generate a smoother flow and convey arguments more clearly.

Lines 303-304: change 'of prevalence of parasites' to 'of the prevalence of parasites' or 'of parasite prevalence'.

Line 316: epidemic should be epidemics

Line 319-320: This example by Ezenwa & Jolles does not fit nicely into the argument being made. Intensity of infection was not tested in this study. Therefore, there is no way to know if the findings of this study agree or disagree with this example.

Line 325-327: Wasn't this parasite also found rarely in this system? It would be nice to add that to help clarify how important this parasite is to this system.

Line 328-329: I understand that this sentence is meant to be a transition sentence, but the connection between prevalence and parasite diversity is not made clearly. Perhaps something like this: "Parasite community composition also effected coinfection prevalence in the mixed effect model." Also, rewrite subsequent parts of this paragraph to make the connection between prevalence and parasite diversity through Simpson's diversity index clearer.

Line 347-350: This sentence is confusing. I am really not sure what it being conveyed. Is it that the ephemeral ponds exist for a limited period of time and the length of that time period can be unpredictable and varied? Or is the purpose to convey something about *D. magna*?

Line 359: change 'relation' to 'relationship'

Line 361-378: Most of the conclusions from this paragraph are drawn from the reverse of the analysis

that was actually run. An analysis addressing the effects of co-infection on host density and perhaps others (looking at morphological signs of infection) could be run. Conclusions from the analyses that were run (starting on line 377) should be focused on unless additional analyses are run.

Lines 374: Please remove 'anyhow'.

Lines 422-435: This is an interesting point. Is it possible that positive effects of co-infections between some parasite species and negative effects of other co-infections could be masking each other to generate the null effect found? Also, is it possible to focus analyses on co-infections with parasites that are known to interact with one another or may be reasonably expected to interact in the host (i.e. gut parasites)?

Lines 437-453: Tie this back into your system/results or remove this paragraph. Is the argument here that if coinfections increase virulence and prevalence drives co-infections than parasites with high overlapping prevalence should be more virulent? Wasn't it stated on lines 432 that the parasites in the gut (most common coinfections) had low virulence? How does this all fit in?

Reviewer #1:

Detailed comments

1.1) There is currently inconsistency in description of the GLMM models. Variables listed in the main text and variables listed in the supplementary table do not match. Supplementary table lists also evenness and richness variables where the main text indicates that only Shannon diversity was used as fixed effect. Hence, it is unclear what models were fitted. The AICc values are also not presented (I expected to find them in the supplementary table of the model variants) despite being an integral part of the analyses and results.

We apologize for the inconsistency and unclarity in the text and materials dealing with the model selection analysis. In the revised version, we took several steps to correct and improve the description and presentation of this analysis. First, following a comment of R3 (see 3.13 below) we removed "time" from the list of tested factors. As part of this change, we also removed all the models with random slope on parasite richness as most of them resulted with a singularity warning. This changed the methods section (lines 408-414) as well as the results (lines 156-163 and table 3) and the discussion (lines 221–233). In addition, we further clarified how different factors were used in the different models by both better stating that while factors from different groups (abiotic and biotic) were used solely in a model, all factors were tested, and by adding a general example (lines 411-413). The supplementary data is now in line with the main text and the AICc values of each model are listed as well the $\Delta AICc$ and W_i values.

1.2) The decision to limit coinfections to 4 parasite species in the null model is not currently explained. I am hence hesitant about whether the null model truly represents a random model. What if the observed maximum of 4 parasites in the data is lower than the null model would suggest if there was no limitation? Wouldn't there then be less coinfections than expected by chance and hence parasite interactions would be interpreted as influential?

When we first drafted the model, it had no limitation on the maximal number of infecting parasites and the results were the same to those we received after adding the limitation. Due to changes after correcting a coding mistake (see answer to comment 1.9), we reran the model both without limitation and with new limitation of maximum 5 species. The results didn't change compared to the previous model nor between the limited and the limitation free models. We further estimate the probability of the bootstrap to draw individuals with high number of parasites (on a non-limited model) and found that the probability to draw individuals with >3 parasite species was low. This strengthens our belief that the limitation does not impact the performance of the model while keeping it more realistic.

1.3) A reader who is not familiar with the *Daphnia* parasite community would greatly benefit of a description of the system in the methods. For example: which parasites are present or expected to be present in the system, what is known about their interactions from lab studies, and how dynamic their presence in the populations are among seasons. These are hinted at (Line 117-118) "the parasites of *Daphnia* have negative impact on each other", but are these the same parasites as in this study? It seems they may not be because one of the parasites is described as "potentially new to science" (Line 247) and

Discussion refers to “the low virulence of the parasites” several times. With more information, it would be easier to take in the Results and Discussion.

We thank the reviewer on this comment. In the revised version, we changed the introduction and included a paragraph that gives a better presentation of the study system for the reader (lines 63-88). We accompanied it with a new table (table 1) that gives the reader specific details on the different parasite species we found in this study. Together these changes present the reader with better information about the host, the different parasite and the nature of their interactions including up to date knowledge on parasite-parasite interaction and coinfection in this system.

1.4) Discussion would benefit from some refocusing and polishing. Currently, the focus seems a little bit of here and there. The paragraph discussing host density in the GLMM model (Line 361-379) seems contradicting to what the model tested for. The paragraph discusses how host density is affected by parasites. However, in the GLMM model host density is used as an explanatory variable to explain parasite coinfections. Hence, the paragraph seems out of scope in the middle of the paragraph while the ending provides some more expected discussion.

Another paragraph that the authors could reconsider is the hypothesizing about virulence evolution (Line 437-Line 454). While I agree that evolutionary effects are important and worthy of mentioning in the manuscript, in its current form and in this specific manuscript, this section seems a bit excessive and out of scope as the study does not specifically address these questions. The paragraph does not seem to discuss a result of this study. In addition, due to the short space provided for the topic, there is no space to discuss the full extent of the questions. Given that the Discussion is already on the longer side, this paragraph could be shortened into one or two sentences and combined to a conclusion paragraph. Or, the point that temporal variation can have evolutionary consequences, could be moved to the section in the Discussion that goes through the temporal patterns in the results.

On the other hand, what I felt was missing is some discussion about what previous studies investigating parasite vs. coinfection prevalence. Is testing this question a neglected topic compared to other mechanisms (host-parasite, and environment drivers) or a frequently addressed topic? Null models have been around for some time (manuscript cites Gotelli 2000) and have also been used to study parasite metacommunity structures also previously (I think at least by Robert Poulin, and Joshua Brian and David Aldridge). Some summary about this would be welcome to help the readers not familiar with the null modeling literature. Similarly, examples about parasite epidemics peaks are surprisingly few in number. If this is because they are rare, it would be worth it to highlight it.

We thank the reviewer on this comment as it helped us to see better the weak spots of the previous version. In the revised version we focused the discussion to deal more with the results and observations that were reported and reduced the topics that were out of scope or more speculative. Specifically, we removed the evolutionary paragraph and added more discussion of previous studies and how it relates to our observations. Additionally, we edited the part discussing the effect of parasites on host density (lines 406-417). It is now accompanied by a new analysis dealing directly with this question, it is better rooted with the literature and provides better interoperating our results.

1.5) Line 29-30 Seems to be missing a word, perhaps “of”.

The word "of" was added (line 23).

1.6) Line 156 How often was this the case? I did not find these data. Is it presented somewhere or could it be added? For example into Table 1.

We appreciate the reviewer comment. Eight samples out of forty-seven samples (from five different populations) that were used in the analysis (numbers within parenthesis in table 2) had less than 100 individuals, however, they varied in their size. Specifically, the number of individuals in each of these samples was as follows: 99, 95, 87, 80, 67, 55, 49 & 38. Each of the four smaller samples belonged to a different population. Therefore, we are certain that these samples do not harm the analysis.

1.7) Line 223 and 244 “For” -> “of”.

Assuming that "244" was 224, both "for"s were replaced with "of" (lines 390-392).

1.8) Line 225 “abiotic variables listed above” It would be helpful to remind the readers what the variables are. If this means the variables listed under “Abiotic parameters”, it does not correspond with supplementary table describing the models where it lists “w. temp”, “time”, and “SPC”. There are also more variables in the supplementary table in the “biotic” models than described in the main text and it is hence difficult to conclude what models were fitted.

We thank the reviewer for this comment. The original line 225 was deleted while changing this part. We corrected the names of the factors in the supplementary data to fit their names in the main text. We also improved the description of the analysis in the methods (lines 408-414) and believe that now this analysis is clearer and easier to understand.

1.9) Line 255 Would it be possible to provide the exact percentage instead of “about 20%”?

We especially thank the reviewer for this comment since it led us to find a coding mistake. The mistake caused a wrong slicing of the data table and one necessary column was left out. We corrected the mistake and found that while the distribution of infection types didn't change (i.e., coinfections were still a relatively small portion from all infections observed), the numbers did. The corrected exact percentages are now presented in the main text (lines 120 and 127-128). Additionally, the maximal number of species infecting an individual changed from four to five (reported in the same place). Therefore, we ran again the null model analysis with the correct maximal number of parasites and the results did not change. Besides these two parts of the analysis, no other analyses were affected by the coding error, since it occurred only in the part of the code relevant to the results discussed here.

1.10) Line 261: relation-> relationship.

The word was replaced (line 126).

1.11) Line 237 I suggest adding the interpretation of this correlation here, which

I suppose it is that the parasite interactions were random. Otherwise readers not familiar with the terminology used in null modelling will be confused to see this term used in the Discussion. It would be good to repeat it in the Discussion when the term first is used.

We thank the reviewer for this suggestion. We added the interpretation (lines 137-138) as recommended.

1.12) Line 285 “The results of the model selection Table 2” I was surprised to not find the AICc comparison results in Table 2 (and later in the supplementary) due to the formulation of this sentence. “The results of the best model based on models selection” or something similar would be more appropriate here.

We thank the reviewer for this comment. The AICc values as well as the Δ AICc and W_i values of all the models are now presented in the supplementary data. Since only one model is presented in Table 3 (previously Table 2), we did not add this value in the main text because it is informative only in the context of the other models. The readers are referred to the supplementary data in the table's caption.

1.13) Line 307 “end of the host activity season” A simple “the end of the season” would be sufficient and more in line with the terms used previously.

The text was changed to "the end of the season" (line 182).

1.14) Line 307 “increasing dominance” If I interpret the results correctly, coinfections only dominate infections in three populations in the end. Perhaps replace with “prevalence”. Also the same situation on Line 257.

"Dominance" was replaced by "prevalence" (lines 121 and 181).

1.15) Line 308-309 Would be helpful to recite which modelling yielded these results.

We appreciate the reviewer's suggestion. However, since these results are discussed in detail in the following discussion where the context of the relevant analyses (null model and network analysis) is further established, we disagree about the potential advantage of mentioning the specific analyses at the introductory phase of the discussion.

1.16) Line 313 Unnecessary words “to some extent”.

These words were deleted as part of the extensive editing of this paragraph.

1.17) Line 313-317 Please cite some examples.

While Wille *et al.* 2015 and Albuixech-Martí *et al.* 2021 were cited on this topic in the original version, we state that the relation of prevalence patterns to coinfection patterns is mostly overlooked (line 187) to the best of our knowledge. Nonetheless, we improved the discussion on the topic and cited several studies that suggest that "overlapping peaks" of prevalence promote coinfection such as Medeiros *et al.* 2016 and Langley & Fairley 1982. These changes appear in lines 186-198.

1.18) Line 322 Unnecessary word “dramatic”.

Deleted.

1.19) Line 327 There seems to be a word missing or some other typo “that its”.
This sentence was deleted as part of the editing process.

1.20) Line 362 Unnecessary word ”very”.
This paragraph was completely replaced.

1.21) Line 374 Suggest changing word ”anyhow” to more formal ”however”.
This paragraph was completely replaced.

1.22) Line 375 ”low-virulent parasites” This would be relevant information to put with a more detailed descriptions about the parasites.
This paragraph was completely replaced. Nonetheless, additional information about the parasites was added in the introduction (lines 63-88) and in table 1.

1.23) Line 383-384 “...was illustrated by an almost perfect match between the identity of the most common coinfection combos and the most prevailing parasite species” This sentence is a little complicated and I had to read it multiple times to understand it. Would there be a more simple way to say this? Also “combinations” instead of “combos” would be more formal.
This sentence was deleted as a part of the editing process of this paragraph.

1.24) Line 395-400 I am a little bit confused as to how the last sentences in this paragraph relate to the beginning of the paragraph. It almost seems as if they have been accidentally relocated to a wrong place in the Discussion. Would it be possible to improve the link and flow somehow?
This part was removed as a part of the editing process of this paragraph.

1.25) Line 410 ”probably” is unnecessary here.
Deleted.

1.26) Line 412 The neutral theory is introduced rather suddenly. If I understand the authors intention correctly, a “however” would be a fitting link word to contrast the idea that all mechanisms need to be taken into account to the idea of neutral theory? Adding a linking word would improve the flow.
We added the word "However" at the beginning of the sentence (line 266).

1.27) Line 417: “For example...r value of ..” This result does not communicate much to a reader who has not read the original paper.
We changed it to "high fit to a neutral model" (line 271).

1.28) Line 428 “Competitive ability is often determined by virulence” Some references of empirical examples would be welcome here.
This part of the discussion was deleted.

1.29) Line 461 “parasites overlap in space is due...” Do you mean time? I would assume time because it was part of the sampling design.

We thank the reviewer for the correction. We change "space" to "time" as originally intended (line 292).

1.30) Figure 3. Could you mention "population" in the first sentence or "Soreq swamp, one of the sampled populations". Because there are 8 network diagrams, it can first seem to be networks of each of the sampled populations instead of temporal points of one, and I got confused at first. It would also be helpful, if the node of the dominant parasite be at fixed position throughout the figures to make it easier to inspect the development of the network. Even better if the parasites in all figures would have fixed positions to enable comparing among populations.

We thank the reviewer on this comment. The phrase "one of the sampled populations" was added after "Soreq swamp". In addition, the locations of all the parasite species are now fixed in all the network figures. However, small changes may occur due to presence\absence of species from the sample (software limitation) but the relative location between the species within a network remains the same.

1.31) Figure 4. Please add to the figure text what analyses these correlations come from.

We added the relevant analysis (network analysis) to the figure's caption.

1.32) Figure S1, S2, “Relation” should be “relationship”.

The word relationship is used in the caption of the new S1. The original S2 was removed from this version due to changes made to this analysis (see details in the

answers to R3).

Reviewer #2 (Remarks to the Author):

Summary: In this manuscript, the authors analyze infection prevalence of each of several parasites infecting ephemeral *Daphnia* populations. They find that rates of coinfection are consistent with a "neutral" model of parasite assembly within hosts, where the likelihood of coinfection of a given set of parasites simply increases with prevalences of the individual parasites in the population. Associations among parasite species were found to be random. The study provides important insights into patterns of coinfection through time in natural populations.

The manuscript is well written overall. I have several comments aimed at increasing the clarity further. My comments are listed in order from start to end of the manuscript. Some particularly important comments are preceded with two asterisks (**).

2.1) L1: I found the start to the title ("May the odds be with you") to be distracting, as it reminds me of famous quotes both from The Hunger Games books ("May the odds be ever in your favor") and from Star Wars ("May the force be with you"). I'm not sure if those references are intended. In any case, I think the title would be better simply as, "Coinfection frequency in water flea populations is a mere reflection of parasite diversity".

We appreciate the reviewer's opinion. Nonetheless, the "Star Wars" reference is most certainly intended as we find it both catches the eyes of the readers and communicates one of our main results. We are delighted to discover that it serves as a reference to the "Hunger Games" in a similar manner. We believe, that the title is the one place where authors can be a bit less scientific and express themselves and live the title as it is.

2.2) L30: Missing "of" in "natural dynamics of coinfection prevalence".

The word "of" was added (line 23).

2.3) L35: Change "bound" to "bounds".

Corrected (line 28).

2.4) L47-48: None of the three papers cited here show that coinfections are common in plants. Please add another citation accordingly.

We thank the reviewer on this comment. We added a citation to Tollenaere et al. 2016 that discusses the implications of coinfections in plants.

2.5) L52: do these "counter examples" show no effect of coinfection on damage to host, or a decreasing effect of coinfection on damage to host? I think the sentence starting at end of L49 should be ended after the citations in L51 and a new sentence started in L52 about the counter examples.

We thank the reviewer on this suggestion. We split the original sentence to two, "Coinfections can affect the amount of damage inflicted on the host, in many cases

increasing it. Yet, other studies demonstrated decreased damage or no changes under coinfections" (lines 41-43).

2.6) L55-58: I found this sentence to be confusing and possibly internally redundant. Could you simplify the start to: "Parasite competition can modify transmission, either directly by altering susceptibility or survival of hosts..."? Actually, the "survival" part of that makes sense to me with regards to modifying transmission, but I'm not sure how "susceptibility" is involved here if the parasites have already co-infected the host. Are you referring to susceptibility to another parasite that hasn't yet had the chance to infect? In the next part of the sentence ("...or indirectly by reducing host availability via a virulent community member...") I cannot tell how "reducing host availability" is different from "altering susceptibility or survival" in the first part of the sentence.

We thank the reviewer for this comment. The previous version contrasted between competition within the host and outside the host, which was out of scope and therefore context. The new version focuses around within-host competition and is better fit with the rest of the text (lines 43-46).

2.7) **L61: Please insert a description of the site of infection within host's body for each parasite. See also my comments below about somehow making this apparent in Fig. 2, S12, and S13. I had to dig through the supplemental data file to find that information in tab called "Parasite species list" (also there is a small typo in header of column D of that data sheet: "witihn").

We thank the reviewer for this comment. The study system including information about the different parasite species is better embedded in the revised manuscript. Details about each parasite, including the site of infection, are now presented in table 1 in the main text. Relevant changes were made to the figures as well and are presented under the relevant comment.

2.8) L166-167: I'm confused by what Figure S1 is showing. I want to think of this dissection vs. prevalence result in terms of being able to detect rates of "false negative" and "false positive" assignments to infection status. But maybe that's not quite right because you don't know "true" infection status for individuals that weren't dissected? Clarification is needed on what we can learn from Fig. S1 in terms of how useful the gut dissections are for detecting gut parasites that wouldn't have been detected if you hadn't performed the dissection.

We thank the reviewer for pointing us the unclarity of this supporting analysis. The goal was to demonstrate that the lower rates of separating the gut during the dissection of *Daphnia* didn't result in lower discovery rate of gut parasites. We would like to clarify here, that every individual was dissected and screened at x400 magnification under phase-contrast microscope. The difference among individuals regarding gut dissection was that we specifically separated the guts from the rest of the *Daphnia* body in some individuals (especially those that lost clarity of the carapace due to infection as explained in lines 343-349). The words "dissected the individual" replaced the words "it was dissected" (line 340) to improve the connection to the first phase of the screening (examination under a stereoscope) and better

inform that all the individuals were treated the same (except for gut separation). The guts and the parasite within them can be seen clearly even when not separating the guts for the experienced observer, especially when no other heavy infection outside of the guts is present, and it is supported by the analysis accompanying figure S12 (previously S1). While the first version correlated between the rate of detection of all the species in the sample combined to the rate of dissection, following additional comment of R2 (see 2.34) we changed the analysis. The new version asks how gut separation during the dissection process is correlated with the detectability of each species. The correlation presents the separation rate on the x axis and the data points are the prevalence (our detection rate) of each parasite species in each sample. The color code of the data points follows the parasite color code in other figures and the reader can see the distribution of each species while the trendline represent the correlation with all of them. We believe that this new version is clearer and better communicating that while acknowledging that false negative could have occurred, it would have been in low rate that is not impairing our "truth" about the presence/absence of gut parasites in our samples.

2.9) Something that would help with interpretation of Fig. S1: Are there multiple points overlapping at for example (0, 0), (0, 1), and (1, 1) positions on the plot? If so, you could scale the size of dots relative to number of overlapping points (e.g., using `geom_count` in `ggplot2` package in R). Also, I suggest changing x-axis titles of Fig. S1 and Fig. S2 to "Proportion of sample dissected" and y-axis titles to "Detected prevalence of gut infection".

We thank the reviewer for these suggestions. In the new version of this figure (now named S12) we allowed the data points to slightly jitter to avoid the problem of overlapping data points. The titles remain the same since they describe properly the data presented. As explained in the answer to comment 2.8, 100% of the individuals were dissected in all the samples. The only thing that varied was the proportion of gut dissection which refers to separating the gut to a different location on the slide, so it

can be observed separately from the rest of the body. Since the data points now present the prevalence (i.e., our detection rate) of each species, we find the axis title a better fit than in the previous version, where the reviewer's suggestion would have fit better. We thank the reviewer for helping us understand how to better communicate our message.

2.10) L203-205: Suggested rewording for clarity: "To improve the assessment of the relation between parasite prevalence and abundance in coinfections, the sum of edge weights connected to a given parasite in every sample was correlated against its respective prevalence."

We thank the reviewer on this comment. We followed the reviewer's suggestions and changed the sentence. The new phrase is "To improve the assessment of the relation between parasite prevalence and their abundance on coinfections, we correlated the sum of edge weights connected to a given parasite in every sample against its respective prevalence" (lines 385-388).

2.11) L212: The term "degree centrality" is defined in the caption of supplemental Fig. S13. Please also insert that definition here in main text.

The definition is now presented in the main text (lines 394-395).

2.12) L226: I assume "time in which the sample was taken" refers to date of year? But could also refer to time of day. Please clarify.

Time is no longer part of this analysis and this line was deleted.

2.13) **L227-229: This is unclear in the "List of GLMM" tab of the supplemental data file. In that data sheet, the abiotic factors listed are "w.temp" (is that an abbreviation for water temperature? Or does the "w" stand for something else?), "SPC" (I know that is specific conductance, but that acronym needs to be spelled out in the table), and "time". Oxygen concentration is not listed as a factor in any of the models in that "List of GLMM" tab. But L229 of main text implies that oxygen concentration was included in some or all models with abiotic factors (unclear).

We apologize for the inconsistency between the supplementary data and the main text. We corrected the supplementary data to be inline with the main text both in models' description and in variables names.

2.14) L256: Suggest deleting "Nonetheless" and simply starting sentence with "The frequency of..."

Deleted.

2.15) L257: Suggest replacing "dominant as" with "common than".

Replaced with "prevalent" (line 121).

2.16) L266: Suggest inserting words "while" and "were" here: "...comprised solely of gut parasite, while coinfections during spring were comprised..."

We took the reviewer's suggestion (lines 131-132).

2.17) L284-286 and Fig. S13: I found this wording to be unclear. Suggest changing from "For a given sample, there was a tendency of normalized centrality to increase with prevalence per species" to "For most parasite species, there was a tendency for normalized centrality to increase with its prevalence in the sample" if that is what you mean.

We took the reviewer's suggestion (lines 151-152).

2.18) L286: Suggest rewording the description of Fig. 4 results to: "However, the overall strength of the positive correlation between prevalence and normalized centrality across all parasite species and samples was relatively low (Pearson's $r=0.41$; Figure 4)..."

We took the reviewer's suggestion (lines 152-153).

2.19) **Discussion: The discussion section is very long. Currently nearly 7 full pages in total. I think two entire paragraphs could potentially be cut altogether, or at least greatly trimmed down: L401-421 (maybe you could weave in the nice references to neutral theory of community ecology elsewhere in Discussion?) and L437-454 (while interesting, the discussion of evolution of parasite virulence seems beyond the scope of this study).

In the revised version of the manuscript, we invested significant efforts in improving the discussion. We improved the focus, fluency and connectivity of the discussion and the changes are present throughout the entire section. The evolutionary paragraph was removed.

2.20) L312: Delete "a" before "strong".

Was deleted as part of the editing process of the discussion.

2.21) L316: Change "epidemic" to "epidemics".

The word "curves" was added after "epidemic" (line 188).

2.22) L319: This is the only place where the term "intensity" is used, so please define here (or replace with "prevalence" if that is what is meant).

Was deleted as part of the editing process of the discussion.

2.23) L324: Suggest starting a new sentence after "... (Figure 1)". For example, "In other populations, coinfection never became the dominant form of infection, possibly due to early extinction of those host populations before the vast proliferation of parasites."

We thank the reviewer for the suggestion. We chose to delete this part of the sentence (lines 193-194).

2.24) L327: Suggest rephrasing to: "...was also the only species with a weak correlation between its prevalence and frequency of coinfections (Figure S12)."

Was deleted as part of the editing process of the discussion.

2.25) L331-333: Suggest rewording to: "The Simpson's diversity index is an

inverse of a dominance index. As such, Simpson's diversity index emphasizes the importance of overlapping prevalence of parasite species rather than their richness per se (Kim et al. 2017). In our study Simpson's diversity index was strongly correlated with time, ... " In the last part of L333 about how this "can explain diversity's high effect size on coinfections" also needs more explanation. Are you trying to point out that from your data set you can't tease apart the effect of Simpson's diversity index from effects of other time-correlated factors?

We thank the reviewer the suggestion and comment. In the revised version, the original paragraph that discussed this result was deleted due to the changes made to the analysis (see answers to comments 3.13) and the editing process. We embedded this result in the discussion of the temporal pattern etc. where it fits better than in a standalone paragraph. The current phrasing is "Moreover, our analysis showed that Simpson's diversity index was a strong predictor of coinfection (Table 3). This diversity index emphasizes prevalence rather than richness per se⁶¹ and further supports the importance of parasite prevalence as a driver of coinfection..." for the new context please see lines 194-198.

2.26) L344: After "additional parameters", consider adding "or additional geographic regions" to link with earlier in paragraph where you discuss limitation of sampling in single region of Israel.

This part was deleted following the changes of the model selection analysis. In short, "time" is no longer a factor in this analysis and the parts discussing it were detected. For more details see answers to comment 3.13.

2.27) L347-350: This sentence should be refined for clarity.

This part was significantly edited. The equivalent for this sentence is now phrased as follows: "Moreover, our analysis showed that Simpson's diversity index was a strong predictor of coinfections (Table 3). This diversity index emphasizes prevalence rather than richness per se⁶¹ and further supports the importance of parasite prevalence as a driver of coinfection" (lines 194-196).

2.28) L384: replace "combos" with "combinations".

Replaced (line 259).

2.29) L391-393: This wasn't highlighted in results figures. But may be apparent if you indicate in figures which of the parasites infect hemolymph.

We acknowledge the reviewer's comment. While this part of the discussion was deleted in the editing process, the revised version of the manuscript is better presenting the infection site of the parasites. These changes are present in the new table 1 that presents the reader with relevant information about the parasite species. Additionally, the term "internal parasites" replaced the term "hemolymph" for the description of this group of parasites, as it better represents it and the variation within it for readers who are not from within the *Daphnia*-parasite world. The division of parasite by general infection site is also present now in all the relevant figures. Please see more details about the graphical changes in the answer to comments 2.33 & 2.34.

2.30) **L437-451: Above, I've suggested cutting this whole paragraph because it goes beyond the scope of the present study and the Discussion section is already quite long. (However, if I wasn't recommending cutting it, I would have commented that many of the ideas in this paragraph would require further explanation and citations. The logic was difficult to follow, particularly for L445-451 where there are no citations provided.)

We thank the reviewer for this comment, this part was deleted from the discussion.

2.31) Table 2: Consider replacing "Sampling Time" with "Sampling date". Also, suggest spelling out "Results of the best generalized linear mixed model (GLMM)..." in caption.

We thank the reviewer on this comment. "Time" is no longer a part of this analysis and therefore omitted from the table. We took the reviewer's suggestion for the change in the caption (table 3).

2.32) **The results of the model competition aren't actually shown in the main text or supplement. Please add the key information (e.g., number of parameters estimated, AICc, delta AICc, Akaike weight, adjusted R²) into the "List of GLMM" tab of the supplemental data file and reference those model competition results in the main text.

We apologize for not implementing this crucial information in the previous version. The AICc, Δ AICc and W_i are now presented in the supplementary data file. We refer the reader to the data in the caption for table 3 and the main text (line 285).

2.33) **Figs. 2, S12, and S13: In the legend, could you please help us keep track of which parasite occupy which site in the host body? This could be done by grouping the parasite names by host site (instead of arranging them alphabetically) with additional labels in legend, or possibly with a shape aesthetic for site in host body.

We thank the reviewer for this suggestion. In the new version of figure 2 (presented below), gut parasites are marked with dashed lines while the internal parasites are marked with solid lines. In figures S10 (presented below) and S11 (previously S12 and S13 respectively) gut parasites are marked with diamonds while internal parasites are marked with circles. Additionally, gut parasites are now marked in a different color in the network figures (e.g., figure 3 present) and their location is fixed to the right side of the network (when internal parasites are not present).

2.34) Figs. S1-S2: Could you label the points by species of gut parasite, using same colour scheme as in Figs. 2, S12, and S13? Can keep the one regression line through all species combined, but it would be nice to see if detection vs. dissection varies among gut parasite species.

We thank the reviewer about this idea as it led to us to improve this analysis. The new version of this figure is presented in the answer to comment 2.8 as well as the changes made in the analyses. In short, the data points of each species are now presented under the same color scheme like in figure 2.

2.35) Figs. 2 and S4: Please add small points to show the actual sampling dates at each site. Or if points are too hard to see amidst all the lines, you could add small tick marks on x-axes of all of these panels to indicate the sampling dates at each site.

Small points were added to mark the actual sampling dates in both figure 2 (presented above in comment 2.33) and S2 (previously S4; presented below at 2.36).

2.36) Fig. S4: The colours are not appearing as described in caption. The "blue" appears black on my screen (maybe it's a very dark blue?) and the "teal" appears as more of a light blue.

We thank the reviewer for pointing it out. We adjusted the colors and their description

in the caption. Coinfection by internal parasites or both by gut and internal parasites are now colored by blue and cyan, respectively.

2.37) Figs. S6-S11: I assume the networks are only drawn for dates where there were at least some (greater than zero) coinfections? This wasn't explicitly stated in the methods.

The assumption is correct and it is now stated in the methods (line 188). In addition, we corrected the tick labels of figure 1. Instead of presenting the name of the month every other tick, month names are now presenting under the tick of the first sample of the given month. This should help the reader to locate samples between figure 1 and the networks.

2.38) Figs. S12 and S13: "Prevalence" is misspelled in x-axis titles of both these figures.

Corrected.

Reviewer #3 (Remarks to the Author):

This paper uses a *Daphnia* model to investigate the potential for co-infections to be driven primarily by parasite prevalence rather than other factors such as interactions between co-infecting parasites. They found that the null model best predicts the frequency of co-infections and that overlapping parasite prevalence in the field directly predicts the likelihood that *Daphnia* would be co-infected. They also found that parasite density and time played a significant role in co-infection prevalence.

I find the investigation and concept of co-infection prevalence as simply a side effect of parasite prevalence interesting and I think they have strong evidence for this concept in their study system. However, their results are partially complicated by the significant effect of time in their model. Time plays a major role in this system, as *Daphnia* populations, parasite populations, and tested abiotic factors were correlated with time. While this is logical in this seasonal system, it makes it somewhat challenging to separate out if the results are driven exclusively by prevalence or also by other abiotic or biotic factors that were not tested. This is something the authors acknowledge. The ephemeral nature of this system makes me question how applicable these results would be to other systems, in which more complex co-infecting interactions may be more likely to develop due to the longer overlap of these parasites in the host system and/or through potential interactions with more complex host immune systems.

They find no effect of interactions between co-infecting parasites in their system. However, they compare co-infections in different parts of the host, which are unlikely to interact at all. Additionally, they acknowledge that the co-infecting parasite interactions that they do not find in their analyses could still be happening through exclusion from the system entirely. I would appreciate a more explicit explanation throughout the manuscript of which parasites or parasite types (gut vs. hemolymph etc) they anticipate interacting and what evidence they found to support that they did not. This is mentioned occasionally throughout the manuscript, but I believe a more thoughtful and thorough approach to this topic could strength the manuscript.

Finally, the paper starts out discussing the importance of co-infections on host health, parasite transmission, etc., but does not address those questions explicitly in the methods or results. There is some evidence from their methods that they could explicitly address these questions if they wanted (see detailed thoughts below). Their most prevalent co-infections included non-virulent parasites, which may mean that, although prevalence is driving these co-infections, these co-infections could play little to no role in individual host health or host populations. Virulence is also frequently brought up in the discussion as being driven by co-infections or driving them, but which of the tested parasites were virulent and how this might be driving co-infections

could be set up in the introduction and connected more thoughtfully in the discussion.

Line by line

Abstract:

3.1) Lines 30-32: This is quite a long sentence. You could add to the flow by breaking it up. I recommend ‘... or simply derive from parasite diversity. Identifying the relative impact...’

We split the sentence as suggested (line25).

3.2) Line 36-37: I find this sentence a bit confusing. I think maybe clarifying the temporal pattern of the co-infection prevalence would help. Something like: “Additionally, coinfection prevalence increased over time, becoming equally common as single infections.”

We changed the phrasing to "...increased over the season..." (line 29).

3.3) Lines 39-40: This statement overreaches the conclusions of your study as you did not address health risks. I think you could say something like: “We suggest that monitoring parasite diversity can help predict where and when coinfection prevalence will be high, potentially leading to increased health risks or increased parasite transmission.”

We thank the reviewer on this comment. We changed the phrasing to the following "We suggest that monitoring parasite diversity can help predict where and when coinfection prevalence will be high, potentially leading to increased health risks to their hosts. (lines 32-34).

Introduction:

3.4) Line 47-48: I would remove ‘including humans’. Co-infections in this system differ in many ways from co-infections in humans.

We thank the reviewer on this suggestion. We kept this phrase since its entire goal is to point out the relevancy of the topic in general and not necessarily to link our system to humans.

3.5) Line 63: For clarity, please change ‘If their effect’ to ‘If the interaction between the parasites’.

We changed the phrasing to " If interactions among the parasites are strong..." (line 51).

3.6) Line 72: I am unsure what is meant by “In return”. I would remove this phrase.

Replaced by correspondingly (line 59).

3.7) Lines 90-92: This means that your co-infection system probably generates different dynamics than longer term systems in which parasites may be forced to interact with one another for growth/reproduction over longer periods of

time. I would not list this among the reasons that this system is ideal for studying co-infections.

We thank the reviewer on this comment. We edited this section of the introduction and in the revised version the part that summarizes the strengths of the system is phrased as follows "Therefore, these populations offer a unique opportunity to study the role of the different drivers of coinfection frequency given their potential for both variation in parasite prevalence and significant interactions among endoparasites" (lines 86-88).

3.8) Lines 94-95: I am curious as to which endoparasitic species reduced each other's fitness. Can you specify the species tested? Also, a basic description of the common parasites or parasite types in daphnia and how you expect them to interact is needed. I think it will set up the hypotheses and results.

We thank the reviewer on this comment. In the revised version we significantly improved the presentation of the study system. The reader is now presented with more details about the parasite species that were found in our populations including which were tested in coinfections and the implications of it. The changes are presented in the introduction (lines 63-88) and in table 1.

Methods:

3.9) Line 128: Add space before citation.

Added.

3.10) Line 157: Here you say that you check for 'morphological signs of infection', but there are no analyses on host health. Could these analyses be added? Understanding how co-infections or single infections altered host health in this study would help bolster the argument made in the first paragraph of the introduction. Co-infections matter when they alter host health or parasite transmission in some way compared to single infections.

We appreciate the reviewer's comment. By "morphological signs of infection" we mean the loss of clarity and the color change inflicted on the *Daphnia* body which give an indication that the individual is infected, sometimes an indication by what and sometimes an indication that the individual is loaded with spores (which we consider when deciding to separate the gut). In other words, it gives some orientation toward the second phase, which is dissection and screening under the phase contrast microscope. While adding an analysis about the effect of parasites on the density of *Daphnia* population (see answer to comment 3.23), it was not possible to achieve any other estimate of host health with the data we collected.

3.11) Lines 225-226: Was parasite prevalence not included in this?

If by "parasite prevalence" the reviewer means to the "general prevalence of parasitism" in the system then the answer is no. However, the prevalence of the different parasite species is embedded in the diversity indices that were used as factors in this analysis. Both Shannon's and Simpson's diversity indices take into account the parasite prevalence of the different species. We believe that using these indices incorporates the effect of parasite prevalence better than simply using the proportion of parasitism since the latter can be high even when no coinfections are present (e.g., one parasite take over the population). The incorporation of prevalence

via these indices also fits nicely with our idea that the overlapping prevalence matters which we discuss in lines 194-198.

3.12) Lines 227-229: It would add clarity to specify which abiotic parameters were correlated with time here. By my understanding only four abiotic parameters were measured, so: “Since pH, water temperature, and conductivity were correlated with time, only one of these abiotic parameters or time was included in a single model along with oxygen concentration.”

We thank the reviewer on this comment. To increase the clarity regarding to how different factors were used in the model we added the following example "e.g., a model from group the last group will contain oxygen concentration, host density, additional abiotic factor (water temperature, specific conductance or PH) and a diversity index (Shannon, Simpson, evenness or richness) as fixed factors" (lines 411-413). We believe that it will help the reader understand which factors were permanent and which alternated in the different groups of models.

3.13) Line 234: The listed models in the supplement do not line up with the description offered in the manuscript. It is stated that only one abiotic variable (SPC, pH, or water temp) or time was included in the model. However, only models with SPC, water temp, or time were included. If those abiotic variables were too correlated to be included in a model together, why run 2 out of the 3 of them separately? Conversely, why not run all three? Additionally, despite explicitly stating on line 229 that oxygen concentration was included in the models, I do not see it included anywhere. Also, I would like some justification included in the manuscript for why specific variables that were correlated were not run in the same model (i.e., no abiotic factors except oxygen and time; line 227), but other correlated factors were (i.e., parasite density and time; line 335). If time is correlated with most other factors and there are no specific hypotheses about time, why is time included in the model?

We apologize for the inconsistency between the supplementary data and the main text. In addition, we thank the reviewer for the comment about the use of "time" as a factor in the model. We accepted this comment and changed the analysis accordingly. In the new analysis "time" is not used as factor in any of the models. Additionally, we dropped the use of "species richness" as a random slope in the analysis, since all the models with random slope in the new analyses resulted in singularity warning. We changed the description in the methods to fit to the change in the analysis (lines 408-414), the new results are presented in table 3 and in lines 156-163 and a new topic of discussion corresponding to the changes in the results is available in lines 221-232. We updated the supplementary data and it is now in line with the main text and present the AICc, Δ AICc and W_i values of all the models.

Results:

3.14) Lines 266-268: This is an interesting temporal result that I do not believe is touched on in this discussion.

In the revised version of the manuscript, we spent significant efforts in improving and shortening the discussion following the comments of the editor and the reviewers. While we agree with the reviewer's impression of this result, this result shares the same characteristics of the temporal pattern of coinfection, i.e., species are likely to

coinfect with the most common species available. Therefore, and unfortunately, we decided not to repeat the same line of interpretation with another result. We do touch it a bit while discussing the effect of parasites and coinfection on host density in lines 281-285.

Discussion:

3.15) Many of the paragraphs in the discussion could be shortened or combined to generate a smoother flow and convey arguments more clearly.

We thank the reviewer on the advice, in the revised version we invested significant effort in improving this section. Out of scope and/or speculative lines of argumentation were removed while we attempted to improve the flow between the different parts and to strength the discussion of the main results via better context to the published literature.

3.16) Lines 303-304: change 'of prevalence of parasites' to 'of the prevalence of parasites' or 'of parasite prevalence'.

Was changed to 'of parasite prevalence' (line 178).

3.17) Line 316: epidemic should be epidemics.

Was changed to 'epidemic curves' (line 188).

3.18) Line 319-320: This example by Ezenwa & Jolles does not fit nicely into the argument being made. Intensity of infection was not tested in this study. Therefore, there is no way to know if the findings of this study agree or disagree with this example.

This presentation of this argument was edited. Now the reference to Ezenwa and Joles is given as an example to the argument that strong negative interactions break disrupt the association between overlapping prevalence and elevated risk for coinfection, which interact better with the results of this study...", albeit strong negative interactions among species are expected to disrupt this association" (lines 197-198).

3.19) Line 325-327: Wasn't this parasite also found rarely in this system? It would be nice to add that to help clarify how important this parasite is to this system.

In fact Mgr was probably the most common parasite we observed (which surprised us since we had no documentation of it so far and we could not cultivate it in the lab). However, this part was removed during the editing process of the discussion.

3.20) Line 328-329: I understand that this sentence is meant to be a transition sentence, but the connection between prevalence and parasite diversity is not made clearly. Perhaps something like this: "Parasite community composition also effected coinfection prevalence in the mixed effect model." Also, rewrite subsequent parts of this paragraph to make the connection between prevalence and parasite diversity through Simpson's diversity index clearer.

We thank the reviewer for these comments. This paragraph was deleted from the

new version of discussion. Instead of "standing alone", this result is now implemented in the paragraph that discusses overlapping prevalence of parasites and the relation to coinfection." Moreover, our analysis showed that Simpson's diversity index was a strong predictor of coinfection (Table 3). This diversity index emphasizes prevalence rather than richness per se⁶¹ and further supports the importance of parasite prevalence as a driver of coinfection". For full context, please see lines 194-198.

3.21) Line 347-350: This sentence is confusing. I am really not sure what it being conveyed. Is it that the ephemeral ponds exist for a limited period of time and the length of that time period can be unpredictable and varied? Or is the purpose to convey something about *D. magna*?

We thank the reviewer for this comment. We improved the phrasing of this section of the discussion and believe that the message is better conveyed. " Our populations developed in an unstable habitat that is difficult to predict, and thus the parasites' window of opportunity is probably narrow. This is because the length of the ponds' hydroperiod is stochastic and so is the persistence of host populations (see variation in the number of sampling events in Table 2; lines 237-238).

3.22) Line 359: change 'relation' to 'relationship'.
Corrected (line 245).

3.23) Line 361-378: Most of the conclusions from this paragraph are drawn from the reverse of the analysis that was actually run. An analysis addressing the effects of co-infection on host density and perhaps others (looking at morphological signs of infection) could be run. Conclusions from the analyses that were run (starting on line 377) should be focused on unless additional analyses are run.

We are thankful for the reviewer for this comment as it led us to add a valuable part to the revised manuscript. We removed this part of the discussion and replaced it with a new analysis about the effect of parasites and coinfection on the density of *Daphnia* populations. The details about the new analysis and its result are presented in lines 165-170, 275-286 and 418-422.

3.24) Lines 374: Please remove 'anyhow'.
Deleted.

3.25) Lines 422-435: This is an interesting point. Is it possible that positive effects of co-infections between some parasite species and negative effects of other co-infections could be masking each other to generate the null effect found? Also, is it possible to focus analyses on co-infections with parasites that are known to interact with one another or may be reasonably expected to interact in the host (i.e. gut parasites)?

We thank the reviewer for these questions. While we removed the paragraph of lines 422-435 from the original version, we do discuss the potential of positive and negative interactions among parasite pairs to cancel each other when taking a community perspective (lines 254-256). We bring examples that similar systems can

end in different results and cite Haukisaalmi & Henttonen, who state this issue as a challenge of revealing parasite interactions via null models.

Unfortunately, we did not have in our data set enough individuals that were coinfecting with a pair of parasites and that the implications of their coinfections were tested in the lab (information about these pairs is available in table 1). It would have been amazing if we had data that would have enabled us to compare interactions from the lab to interactions from the field. In addition, from the knowledge we possess, there is no reason to assume that some parasites are more likely to interact since similar implications for coinfections were observed regardless of the site of infection (see table 1 and lines 70-81). Moreover, we played with different subsets of the data, but we didn't receive any different results.

3.26) Lines 437-453: Tie this back into your system/results or remove this paragraph. Is the argument here that if coinfections increase virulence and prevalence drives co-infections than parasites with high overlapping prevalence should be more virulent? Wasn't it stated on lines 432 that the parasites in the gut (most common coinfections) had low virulence? How does this all fit in?

We thank the reviewer for this comment. This paragraph was entirely removed from the revised manuscript.

Reviewers' comments:

Reviewer #1 (Remarks to the Author):

This is the second time I have read the manuscript by Halle et al. where *Daphnia magna* populations were sampled through time for parasites to see how parasite and coinfection prevalences develop, which parasites form coinfections, and which other conditions affect coinfection prevalence. I was delighted to read this manuscript again as I continue to see the study well set up and the results interesting. I especially thank the authors for the additional information about the host-parasite system (Table 1 especially) and fixing the GLMM model descriptions. The discussion has been heavily edited and it has shortened significantly. I find that the focus and flow of the Discussion have improved but there is still some need to improve the logic and balance so that the key questions introduced in the beginning of the manuscript are highlighted and the content better reflects the beginning of the manuscript. There are some sections that could be condensed, specifically where results from other studies are discussed, to make space to elaborate on some results that are now left with less attention. Please see the comments 3 and 4 as well as line-by-line comments for suggestions. My main concerns lie with the added analyses on how infections affect host density. Disentangling a reciprocal, and potentially environmentally and temporally structured relationship such as host-parasite population size dynamics, especially with multiple parasites, is contextually complex. More information about the decisions behind the models are needed, especially why abiotic variables are not included. A plot of host density development would help interpret the relationship between host density and parasites especially since the relationship could be reciprocal. The research question is also not mentioned in the beginning of the manuscript. Comment 1 goes through the concerns.

1. My main concerns lie with the additional analyses on how infections affect host density. Disentangling a reciprocal, and environmentally and temporally structured relationship such as host-parasite population size dynamics, especially with multiple parasites is contextually complex. A plot of host density development over time would help understand the seasonal patterns of the populations in this system. The question should also be added into the Introduction to the list of research questions with brief introduction of why it was tested for. Since no abiotic variables were included, some explanation on their exclusion is necessary. Are *D. magna* populations not dependent on abiotic conditions? Plots of the modelling results would help comprehend the results.

2. In the revised manuscript, there is no temporal structure in the GLMM models. The Methods mention that all variables (but not O2) correlated with time. It makes it difficult to tease apart whether the observations are driven by the variables or time. This should somehow be addressed when discussing results such as: "In our study, water temperature was found as an important predictor of coinfection (Table 3)." L224

3. There is still some need to improve the logic and balance of the Discussion so that the content better reflects the key questions introduced in the beginning of the manuscript (Title, Abstract, Introduction). The final paragraph of the Introduction nicely and clearly lays out the questions and predictions about the coinfection network analysis but the discussion is highlighting the effect of temperature and seasonality more. Similarly, the result about the diversity index is mentioned in a paragraph about temperature although the title of the manuscript and the key conclusion in the abstract suggest that the relationship between diversity and coinfections is the most important result. Modifying the Discussion so that the diversity and coinfection network results are elaborated on and the predictions in the Introduction are revisited would better reflect the beginning of the manuscript and strengthen the narrative.

4. The null-models are currently discussed in two paragraphs with the network results in between and it is difficult to follow. The main null-model result of this study (coinfection prevalence follows the expectation of the model, L247-248) is followed by discussion of null-models that simulated coinfection pairs. I find this confusing because this study only used a null-model to look at coinfection prevalence, not the identity of the coinfecting species (L247-261). It would be more logical to first discuss the prevalence result and the neutral theory (currently the topic of the next paragraph starting L262) and then continue on to the networks

Line-by-line

5. L89 a typo in "we combining"

6. L120 suggest changing the sentence into "We found coinfection in 33.28% of the infected *D. magna* individuals."

7. L157 using some other separator than "," would be clearer

8. L165 "parasite infection affected..." do you mean "parasite prevalence"?

9. L187 "Parasite species with overlapping epidemic curves have a higher chance to become involved in coinfections than temporally segregated parasites, especially when they reach their peak prevalence at a similar time. Overlapping prevalence was correlated with elevated levels of coinfections in natural populations of mallards and common cockles 33,37." This is unnecessary repetition and therefore I suggest deleting the second sentence and placing the refs into the more generalized first statement

10. L195 add species to "species richness"

11. L198 "Temporality is a strong characteristic of our study system, extending from the ephemeral nature of the habitats through the appearance and disappearance of *D. magna* populations" This sentence is a bit unclear; specifically what is "extending" and "through" what

12. Line 202 suggest changing to "including parasites" from "specifically"

13. L204-212 Here it seems like results of another study are discussed. I am also not sure why vectors are focused on so heavily here. It is a bit confusing since *Daphnia* parasites are usually not vectored if I remember correctly? For example, there are many plant-fungus epidemics that also follow seasonal patterns and are restricted by off-seasons or crop harvesting

14. L211 unclear what "benefit" refers to here

15. L212-214 "Additionally high turnover rate of the host can form strong seasonal patterns even for hosts with more complex immune response or long-lasting parasites 61,62." The connection from the previous sentence to this is not clear. Therefore it is confusing to suddenly read about turn over rate

16. L214-216 "For example, the prevalence of wood mice parasites changes along the season and though overlooked, coinfections are unavoidable due to overlapping prevalence of different species 61." The term "overlook" can sound a bit negative or criticizing so I suggest changing it to: "and though not tested specifically". Or just delete this sentence and only keep the sentence before this one that gives this same statement more generally

17. L216-218 It is not clear how these statements relate to the this study, some linking by mentioning what this system is about could help. In the sentence, replacing "These" with "the two types of parasites", removing "most" before "most likely" and "this encounter" with "their interaction"

18. L218 unclear what "these" refers to

19. L220 Suggest removing "probably" to something like: "Changing temperature is often a core feature of seasonality."

20. L233 "Kirk et al. 69 demonstrated that water temperature can drive epidemics in *D. magna*..." Because the first sentence of the paragraph is about temperature, it seems that this paragraph is continuing to discuss the effects of temperature. Then it moves on to seasonality, which was also covered in an earlier paragraph. The authors could consider moving some of the contents to the previous paragraphs

21. L235 change "hist" to "host"

22. L248-250 "We found that the observed prevalence of coinfections did not deviate from the predictions of our null-model (Figure S4). Null models were used to estimate parasite associations in

various of host-parasite systems 80–82, yielding both random and non-random associations.”
Discussing null-models that test for the identity of the coinfecting species is confusing because the null model in this study was not used to simulate co-infection pairs and it only tested for prevalence of coinfections

23. L249: delete “of”

24. L250-252 “Interestingly, even similar systems can result in different associations. For example, species of reptile malaria were found to assimilate coinfections randomly in Anolis lizards in one case 83 and non-random, probably due to negative associations, in another 81” I agree that this is fascinating. But I still have to suggest cutting these sentences because they distract from the discussion of the results of this study. Reference 83 could be placed into the previous sentence

25. L254-256 suggest changing “altogether” to “as a whole”, “ended in” to “resulted in”, and “cancel each other” to “cancel each other out”

26. L256 “accomplished” should be “accompanied”

27. L258 unclear to what “their impact” refers to

28. L262-274 To improve the flow and to avoid thematic repetition, I suggest moving this paragraph before the previous paragraph and including the prevalence result here

29. L275 “host population density”

30. L277 Please mention that this is about *D. magna* fecundity. Somewhere here it would be appropriate to refer to the existing literature about the detrimental effects of coinfection more broadly than just *D. magna* studies

31. L278-1279 “It is possible that host density in our populations is regulated through the increasing effect of parasites on host fecundity. This can explain why coinfection yielded larger estimate in the model than single infection, despite having a similar effect size (-0.35 and -0.33 for single infection and coinfection, respectively).” Especially these types of statements make the reader want to see the plots of host density development and modelling results

32. L282: “been” should be “being”

33. L285-286 “Additionally, under natural conditions these parasites probably bare more costs than can be observed in the lab.” Is there some reason or previous reference to this assumption?

34. L363-364 I appreciate the explanation that the restricted model was used to compare the result before selecting to present results from the restricted model. However, I do not see it mentioned in the manuscript yet

35. L363 add “the” before model’s

36. L398: “theme” should be “them”

37. L411 delete the extra “group” in the sentence

38. L408-411 Please explain here that time was not included as a variable

39. Table 1. Thank you for including Table 1, it gives a clear and comprehensive description of the system. Do the parasites transmit similarly or differently? If their transmission is different, it could be added into the column “Category” which could be replaced with “infection site/transmission”

Reviewer #2 (Remarks to the Author):

Line 44 (“...on their reproduction”): replace “their” with either “host” or “parasite”. Currently unclear which is being referred to. Effects on either host or parasite reproduction could affect disease transmission.

Figure S12: “dissection” is misspelled on x-axis

Line 196: insert comma before “albeit”

Line 206-207: "Similar to our findings, the two parasites had overlapping prevalence pattern in both host and vectors and coinfections were present." Change to "patterns." Also, did that study indicate how prevalent coinfections were? That coinfections were "present" doesn't give us much information, so perhaps you could tell us if coinfections were common or not.

Line 214: change to "immune responses"

Line 216: change to "different parasite species"

Line 216-217: This point isn't clear. Why would seasonal parasites with short parasitemia be especially likely to encounter long-lasting parasites? Do you mean relative to parasites with short parasitemia encountering other parasites with short parasitemia? (Because long-lasting parasites should also be likely to encounter other long-lasting parasites, right?) If I am understanding your intent correctly, you could rephrase as: "Seasonal parasites with short parasitemia may be especially likely to encounter long-lasting parasites compared to other parasites with short parasitemia. The implications of encounter between parasites with different *infection durations (*or "life histories"?) is worth future inquiry." (Note also that I suggest deleting the "Another interesting direction is related to the interaction between..." prelude to the sentence.)

Line 220: you can delete the word "probably"

Line 222: Simplify to: "In our seasonal system, temperature plays..."

Line 224: Rephrase as "We found water temperature to be an important predictor..."

Line 227: Citation needed at end of sentence: "...likelihood of infection (citation needed here)."

Line 227-228: Is the ambiguity in this sentence ("seem to"...."exceptions have been reported") reflecting methodological differences between studies, differences between parasite species, differences between provenance of particular parasite species (e.g., parasite populations could be locally adapted to different environments, and therefore have different thermal performance responses), or different host species contexts (e.g., Gh x E or Gh x Gp x E interactions)? Please clarify which of those factors contribute to differences in parasite thermal performance alluded to in this sentence.

Line 235: "host" is misspelled

Line 239: change "setup" to simply "set"

Line 247: Can delete "We found that" and start sentence with "The observed prevalence..."

Line 248: no hyphen needed in null model

Line 249: delete "of" to change to: "various host-parasite systems"

Line 251: what does "assimilate" mean in this instance? Do you mean "acquire"?

Line 252: change to "non-randomly"

Line 255: change "ended in" to "yielded" or "resulted in"

Line 256: With the network analysis, are you actually "eluding associations"? Or "resolving" them? Or "accounting for" them?

Line 286: replace "bare" with "bear"

Reviewer #3 (Remarks to the Author):

Overall, I appreciate the efforts taken to revise this manuscript. The authors addressed many of my comments.

However, there is still a disconnect between the focus of the introduction and that of the discussion. The introduction largely focuses on parasite interactions and how they affect host health. Whereas the discussion largely focuses on the temporal nature of the system. I think the discussion could be improved by a paragraph explicitly discussing parasite-parasite competition/interactions in the host and speculation as to why they might not have occurred/been important in this system.

Alternatively, if the authors do not feel like there is enough information from their study and outside sources to address these questions in this system, then the focus of the introduction could be shifted away from parasite interactions and be focused more on the topics covered in the discussion. Such as the temporal nature of this host-parasite system or the neutral theory of community ecology.

More minor line-by-line comments are below:

Line 43: There is an extra space between 'survival' and 'can'.

Lines 36-46: I appreciate that effort was put into this section but I still think it could use work. Overall, I think this paragraph should be reassessed for flow and purpose. Specifically, this paragraph covers parasite-parasite interactions and the effects of coinfections on host health. I find that the transition between these two topics is a bit confusing and that in some places it's hard to keep track of whether the host or parasite is being discussed. For example, the two sentences on lines 40-42 are discussing damage that the co-infections cause to the host, but the sentence starting with 'the effect of...' abruptly switches to host survival rather than host damage and also discusses parasite competition. Some transition there would be appreciated. Additionally, on line 43 when 'modified transmission rate' is discussed, I believe this would be clearer if it was written 'modified parasite transmission rate'. Finally, the sentence starting on line 44 is particularly confusing. As both hosts and parasites reproduce it would be nice to have some clarity as to which organisms' reproduction is being affected.

Line 57: I'm not sure 'correspondingly' is the correct word to use in this context. I would remove.

Line 72: Suggest changing 'several studies examined' to 'several studies have examined'.

Line 75: This sentence is confusing. I think because 'develop an infection' is a phase usually associated with a host and not a parasite. Suggested rewording: 'Within-host competition between differently virulent parasites can limit the ability of the less virulent parasite to successfully infect the host'.

Line 78: Same issue with 'form an infection' as described above.

Line 89: Suggest changing 'combining' to 'combined'.

Line 157: Suggest change from "that affect the coinfection rate, Simpson's" to "that affect coinfection rate: Simpson's".

Line 194-197: Long sentence. Consider breaking it up.

Line 214: Suggest changing 'response' to 'responses'.

Line 215: I'm not sure what the purpose of the phrase 'though overlooked' is in this sentence. Are you saying that nobody has studied coinfections in this system?

Your discussion devotes 4 paragraphs (over 2 pages of text) to seasonality. While I agree this is an important part of your study and should be well addressed, this section could be reduced. As an example, I was fond of the paragraph in lines 220-231. It was concise and the arguments/connection to your own work was clear. Additionally, perhaps if this seasonality section was reduced, some time could be spent on how highly temporally dependent host-parasite systems might generate different within-host interactions than would be seen in longer-term systems.

Line 235: I am not sure what 'hist density' is. Host density?

Line 248: When you say 'null models were used' this suggests that you ran these models. However, as you have citations, I believe this phrase refers to models that other people have run. Suggest rewording to 'null models have been used' to clarify that this is in reference to previous work.

Line 258: What does 'e.g.' refer to? Was there supposed to be an example here? If you're referring to the citation, I think you can just remove the e.g.

Line 257: Suggest changing to 'revealed a strong correlation'.

Line 285: If these parasites 'probably bare more costs' it would be nice to write why. If it is not probable or challenging to speculate on why you could change this to 'may bare more costs'.

Line 349: I think the figure you label here as figure s1 is now figure s12.

Line 381-382: Consider rephrasing/reassessing grammar in the sentence starting with "we calculated the...". I'm not following.

Table 1: I appreciate the inclusion of this table but it is really clunky. It looks like some species names are bolded and some are not. The word 'amoeba' in the group column appears to be a different font than the rest of the table. The figure legend is missing an ending period. I would also recommend widening the 'changes under coinfections' column to reduce table size.

Supplementary data: This needs another look over for spelling. On the 'List of GLMM final' sheet oxygen is spelled wrong every time and diversity is spelled wrong in model 6. However, I appreciate all the changes made to this. It is much clearer and matches the manuscript.

Figures S10 & S11. I think changing the line type vs. dot shape (as you do in Figure 2) for the different parasite groups may generate clearer definition between the groups. If it is still challenging to see differences in line types (dashed vs. not) when the lines overlap, perhaps you can try both dots and lines would be the clearest.

Reviewer #1 (Remarks to the Author):

This is the second time I have read the manuscript by Halle et al. where *Daphnia magna* populations were sampled through time for parasites to see how parasite and coinfection prevalences develop, which parasites form coinfections, and which other conditions affect coinfection prevalence. I was delighted to read this manuscript again as I continue to see the study well set up and the results interesting. I especially thank the authors for the additional information about the host-parasite system (Table 1 especially) and fixing the GLMM model descriptions. The discussion has been heavily edited and it has shortened significantly. I find that the focus and flow of the discussion have improved but there is still some need to improve the logic and balance so that the key questions introduced in the beginning of the manuscript are highlighted and the content better reflects the beginning of the manuscript. There are some sections that could be condensed, specifically where results from other studies are discussed, to make space to elaborate on some results that are now left with less attention. Please see the comments 3 and 4 as well as line-by-line comments for suggestions. My main concerns lie with the added analyses on how infections affect host density. Disentangling a reciprocal, and potentially environmentally and temporally structured relationship such as host-parasite population size dynamics, especially with multiple parasites, is contextually complex. More information about the decisions behind the models are needed, especially why abiotic variables are not included. A plot of host density development would help interpret the relationship between host density and parasites especially since the relationship could be reciprocal. The research question is also not mentioned in the beginning of the manuscript. Comment 1 goes through the concerns.

1.1 My main concerns lie with the additional analyses on how infections affect host density. Disentangling a reciprocal, and environmentally and temporally structured relationship such as host-parasite population size dynamics, especially with multiple parasites is contextually complex. A plot of host density development over time would help understand the seasonal patterns of the populations in this system. The question should also be added into the Introduction to the list of research questions with brief introduction of why it was tested for. Since no abiotic variables were included, some explanation on their exclusion is necessary. Are *D. magna* populations not dependent on abiotic conditions? Plots of the modelling results would help comprehend the results.

We thank the reviewer for this comment. We added a description of the research questions related to this analysis to the relevant part of the Introduction (lines 108-115). In addition, we tested for the potential of each of the abiotic parameters using LMM. All the models turned insignificant and their details along with the details of the other models (related to the parasites) were added to the Supplementary Data. We tested the parameters in a single model due to their correlation with time. This is discussed in the Methods, where we present the model selection analysis. While using a similar logic for this analysis, we chose to run oxygen concentration separately from the other parameters, in order to simplify the analysis, as in case it was significant, a more complex model would have been considered. We discuss this logic in the relevant part of the Methods (lines 450-452). While we acknowledge the potential effects abiotic parameters can have on *Daphnia* populations, it is not within the scope of this study. Therefore, we added the additional models only to make sure we do not miss any effect that we should have considered when interpreting the results. However, since all the abiotic parameters had no effect, we mentioned it in the Results section, but do not discuss it any further. Attached below are, plots of the parasite related models that were added to the Supplementary Figures along with a density development figure.

1.2. In the revised manuscript, there is no temporal structure in the GLMM models. The Methods mention that all variables (but not O2) correlated with time. It makes it difficult to tease apart whether the observations are driven by the variables or time. This should somehow be addressed when discussing results such as: “In our study, water temperature was found as an important predictor of coinfection (Table 3).” L224

We appreciate the reviewer's comment and concerns regarding the removal of 'time' as a predictor in our analysis. While 'time' was used as a predictor in this analysis, we were advised (in the previous revision) to remove it, since it seems we have no specific hypothesis regarding time. We revised our own list of hypotheses to find this advice to have a strong point and we believe that the removal of 'time' improved the analysis and the interpretation of our observation. In our view, the model selection helped us to identify the most influential parameters from those that 'compose time'. This is because 'time' in our case is not a driver, but rather an axis that different parameters change along with its progression.

1.3. There is still some need to improve the logic and balance of the Discussion so that the content better reflects the key questions introduced in the beginning of the manuscript (Title, Abstract, Introduction). The final paragraph of the Introduction nicely and clearly lays out the questions and predictions about the coinfection network analysis but the discussion is highlighting the effect of temperature and seasonality more. Similarly, the result about the diversity index is mentioned in a paragraph about temperature although the title of the manuscript and the key conclusion in the abstract suggest that the relationship between diversity and coinfections is the most important result. Modifying the Discussion so that the diversity and coinfection network results are elaborated on and the predictions in the Introduction are revisited would better reflect the beginning of the manuscript and strengthen the narrative.

We appreciate the reviewer's concerns. As stated by the reviewer, we made predictions regarding the link between the null model and the dominance of parasite-parasite interactions. We also stated that the goal of the network analysis is to further identify the presence of parasite-parasite interactions. Both topics are now covered in the Discussion and address the research question. Additionally, in the beginning of the last paragraph, we stated the goal and prediction regarding temporal patterns of coinfections. This topic is covered in the Introduction, by elaborating on the relation to parasite dynamics as an alternative to parasite introduction and the relation to environmental factors that contribute to this pattern, both in the short- (our study) and long-term (the paragraph regarding habitat instability). Although the final paragraph of the Introduction dedicates more text to the null model, it is not because temporal patterns are less important to the study, but because more types of analysis are dedicated to it and two predictions were made.

We implemented some textual changes to the Discussion, in order to improve the balance between the two topics our study is focused about. Specifically, we added a paragraph (lines 272-299) discussing possible reasons for "why the interactions found in the lab are not reflected in our observations" and modified other parts of the text to better highlight the connection between our study and the Discussion.

Regarding the comment on the discussion of the effect of parasite diversity, the entire paragraph in which the effect of Simpson's diversity index on the likelihood of coinfection is discussing the same result. As the diversity of parasites consists of both richness and prevalence, we don't distinguish between arguments regarding parasite prevalence and overlapping peaks of prevalence among parasite species (the embodiment of changes in diversity through time in our system and other examples presented therein), and arguments that explicitly use the term 'diversity'. The results from the model are simply supporting what is well pronounced in Figures 1 (prevalence of coinfections increase through time) and 2 (the prevalence of multiple parasite species, in other words parasite diversity, increases in parallel with the increase of coinfections), and we believe that is serving this cause well. We would have changed the word 'diversity' in the title to 'prevalence' if we thought it would serve the paper, however, 'parasite prevalence' can be understood as the general level of parasitism, which can be identical if caused by a single or multiple parasites. Since diversity better reflects the presence of multiple parasites, we choose to stick to that term. We therefore made small modification to this paragraph to communicate our thoughts more clearly.

1.4. The null-models are currently discussed in two paragraphs with the network results in between and it is difficult to follow. The main null-model result of this study (coinfection prevalence follows the expectation of the model, L247-248) is followed by discussion of null-models that simulated coinfection pairs. I find this confusing because this study only used a null-model to look at coinfection prevalence, not the identity of the coinfecting species (L247-261). It would be more logical to first discuss the prevalence result and the neutral theory (currently the topic of the next paragraph starting L262) and then continue on to the networks.

We thank the reviewer on this comment. We reordered the paragraphs and now they appear in the order suggested by the reviewer. (lines 248-271)

Line-by-line

1.5. L89 A typo in “we combining.”

Corrected. (line 89)

1.6. L120 Suggest changing the sentence into “We found coinfection in 33.28% of the infected *D. magna* individuals.”

We accepted the reviewer's suggestion. (line 127)

1.7. L157 Using some other separator than “,” would be clearer.

We replaced “,” with “:” (line 163)

1.8. L165 “Parasite infection affected...” Do you mean “parasite prevalence”?

We thank the reviewer for this comment. 'Infection' was replaced with prevalence. (line 172)

1.9. L187 “Parasite species with overlapping epidemic curves have a higher chance to become involved in coinfections than temporally segregated parasites, especially when they reach their peak prevalence at a similar time. Overlapping prevalence was correlated with elevated levels of coinfections in natural populations of mallards and common cockles 33,37.” This is unnecessary repetition and therefore I suggest deleting the second sentence and placing the refs into the more generalized first statement.

We implemented the reviewer's suggestion. (line 202)

1.10. L195 Add species to “species richness.”

Corrected. (line 206)

1.11. L198 “Temporality is a strong characteristic of our study system, extending from the ephemeral nature of the habitats through the appearance and disappearance of *D. magna* populations” This sentence is a bit unclear; specifically what is “extending” and “through” what?

This sentence was removed.

1.12. Line 202 suggest changing to “including parasites” from “specifically.”

We took the reviewer's suggestion. (line 211)

1.13. L204-212 Here it seems like results of another study are discussed. I am also not sure why vectors are focused on so heavily here. It is a bit confusing since *Daphnia* parasites are usually not vectored if I remember correctly? For example, there are many plant-fungus epidemics that also follow seasonal patterns and are restricted by off-seasons or crop harvesting.

We appreciate the reviewer's comment. This paragraph was edited and shortened and this example has a smaller share of the text in the new version (lines 209-224). We disagree with

the reviewer's opinion regarding our usage of vector-borne systems as examples. First, it helps to demonstrate that our results are relevant to different and diverse types of systems (an objective that increases the target audience of the paper and one that we were specifically asked to improve by the reviewers and the editor in the previous revision). Second, we have additional examples from non-vectorized systems.

1.14. L211 Unclear what “benefit” refers to here.

We thank the reviewer for this comment. The text was modified during the editing process of this paragraph. Specifically, this line was replaced with "Similar systems also have the potential to demonstrate a strong link between prevalence patterns and coinfection,..." (lines 215-218)

1.15. L212-214 “Additionally high turnover rate of the host can form strong seasonal patterns even for hosts with more complex immune response or long-lasting parasites 61,62.” The connection from the previous sentence to this is not clear. Therefore it is confusing to suddenly read about turnover rate.

We appreciate the reviewer's comment. However, this sentence is not supposed to be connected to the previous one, but to the general topic of the paragraph that discusses seasonal patterns and their relation to coinfection in other study systems. Hence, we believe it is well in context with the rest of the paragraph.

1.16. L214-216 “For example, the prevalence of wood mice parasites changes along the season and though overlooked, coinfections are unavoidable due to overlapping prevalence of different species 61.” The term “overlook” can sound a bit negative or criticizing so I suggest changing it to: “and though not tested specifically”. Or just delete this sentence and only keep the sentence before this one that gives this same statement more generally.

Thanks, we replaced “overlooked” with "not tested." (lines 220-221)

1.17. L216-218 It is not clear how these statements relate to this study, some linking by mentioning what this system is about could help. In the sentence, replacing “These” with “the two types of parasites”, removing “most” before “most likely” and “this encounter” with “their interaction.”

We thank the reviewer for this comment. We removed this statement from the revised version.

1.18. L218 unclear what “these” refers to.

Please see our response to the previous comment.

1.19. L220 Suggest removing “probably” to something like: “Changing temperature is often a core feature of seasonality.”

The word "probably" was removed.

1.20. L233 “Kirk et al. 69 demonstrated that water temperature can drive epidemics in D. magna...” Because the first sentence of the paragraph is about temperature, it seems that this paragraph is continuing to discuss the effects of temperature. Then it moves on to seasonality, which was also covered in an earlier paragraph. The authors could consider moving some of the contents to the previous paragraphs.

We removed this sentence from the paragraph.

1.21. L235 Change “hist” to ”host.”

Corrected. (line 238)

1.22. L248-250 “We found that the observed prevalence of coinfections did not deviate from the predictions of our null-model (Figure S4). Null models were used to estimate parasite associations in various of host-parasite systems 80–82, yielding both random and non-random associations.” Discussing null-models that test for the identity of the coinfecting species is confusing because the null model in this study was not used to simulate co-infection pairs and it only tested for prevalence of coinfections.

We thank the reviewer for pointing on this point of confusion. We changed the text and replaced 'parasite associations' with 'coinfections' and the end of the sentence was changed to 'likelihood of coinfections' instead of 'associations'. We also edited the following sentence in a similar manner resulting in a jargon that is a better fit to the context of other parts in the text. (lines 260-262)

1.23. L249: delete “of.”

Deleted.

1.24. L250-252 “Interestingly, even similar systems can result in different associations. For example, species of reptile malaria were found to assimilate coinfections randomly in Anolis lizards in one case 83 and non-random, probably due to negative associations, in another 81” I agree that this is fascinating. But I still have to suggest cutting these sentences because they distract from the discussion of the results of this study. Reference 83 could be placed into the previous sentence.

Done as per reviewer’s suggestion. (line 262)

1.25. L254-256 Suggest changing “altogether” to “as a whole”, “ended in” to “resulted in”, and “cancel each other” to “cancel each other out.”

Done as per reviewer's suggestions. (lines 264-266)

1.26. L256 “accomplished” should be “accompanied.”

Corrected. (line 266)

1.27. L258 Unclear to what “their impact” refers to.

We thank the reviewer for this comment. We replaced this part with "the structure of" to

improve the connectivity within the sentence and to clarify better the result it discusses. (line 268)

1.28. L262-274 To improve the flow and to avoid thematic repetition, I suggest moving this paragraph before the previous paragraph and including the prevalence result here.

Done as per reviewer's suggestions.

1.29. L275 “host population density.”

Corrected. (lines 300-301)

1.30. L277 Please mention that this is about *D. magna* fecundity. Somewhere here it would be appropriate to refer to the existing literature about the detrimental effects of coinfection more broadly than just *D. magna* studies.

The word 'individuals' was replaced with '*D. magna*' to indicate the organism studied by Stirnadel and Ebert. We are now citing studies of non-*Daphnia* systems that demonstrate detrimental effects of coinfections to the host. (lines 302-304)

1.31. L278-1279 “It is possible that host density in our populations is regulated through the increasing effect of parasites on host fecundity. This can explain why coinfection yielded larger estimate in the model than single infection, despite having a similar effect size (-0.35 and -0.33 for single infection and coinfection, respectively).” Especially these types of statements make the reader want to see the plots of host density development and modelling results.

We thank the reviewer for this comment. We added plots of the regression models and of the host density development along with the development of infections by single or multiple species to the supplementary figures file (See Figures S12-14).

1.32. L282: “been” should be “being.”

Corrected. (line 310)

1.33. L285-286 “Additionally, under natural conditions these parasites probably bare more costs than can be observed in the lab.” Is there some reason or previous reference to this assumption?

We thank the reviewer for this comment. We replaced "probably" with "may" to form a more cautious version of this statement. (line 313)

1.34. L363-364 I appreciate the explanation that the restricted model was used to compare the result before selecting to present results from the restricted model. However, I do not see it mentioned in the manuscript yet.

We added to the relevant part in the Results the following comment "a preliminary test of a

model without limitation on the maximal number of infecting parasites yielded similar results." (lines 142-143)

1.35. L363 add "the" before model's.

Corrected. (line 390)

1.36. L398: "theme" should be "them."

Corrected. (line 423)

1.37. L411 Delete the extra "group" in the sentence.

Deleted.

1.38. L408-411 Please explain here that time was not included as a variable.

We thank the reviewer for this comment. All variables used in the different models of this analysis are detailed in the example given several lines below (lines 437-439). We think that understanding that 'time' is not one of these variables is a fair conclusion and an additional statement will be unnecessary repetitive.

1.39. Table 1. Thank you for including Table 1, it gives a clear and comprehensive description of the system. Do the parasites transmit similarly or differently? If their transmission is different, it could be added into the column "Category" which could be replaced with "infection site/transmission."

In general, most of the known *D. magna* parasites transmit horizontally. Few microsporidia such as *H. tvaerminnensis* use mixed-mode transmission, and can be transmitted vertically in addition to horizontal transmission. In our study, besides *H. tvaerminnensis*, all the other species are known or assumed (unknown or unstudied species) to use only horizontal transmission. We therefore added a comment below Table 1 that explains the difference in transmission between *H. tvaerminnensis* and the other species. Furthermore, in the table we explain how host and parasite fitness are affected by transmission mode.

Reviewer #2 (Remarks to the Author):

2.1. Line 44 ("...on their reproduction"): replace "their" with either "host" or "parasite". Currently unclear which is being referred to. Effects on either host or parasite reproduction could affect disease transmission.

Corrected. (line 44)

2.2 Figure S12: "Dissection" is misspelled on x-axis.

Corrected.

2.3. Line 196: Insert comma before "albeit."

Corrected.

2.4. Line 206-207: "Similar to our findings, the two parasites had overlapping prevalence pattern in both host and vectors and coinfections were present." Change to "patterns." Also, did that study indicate how prevalent coinfections were? That coinfections were "present" doesn't give us much information, so perhaps you could tell us if coinfections were common or not.

We thank the reviewer for this comment. We corrected 'pattern' to 'patterns' (lines 212). As for the additional info regarding coinfection in this study, the percentage of coinfections in the robins is not stated in the paper and for the mosquitos the probability of having one parasite is higher if the other parasite is present, but data of coinfections in the samples is absent. So unfortunately stating anything more indicative about coinfections from this study would be speculative.

2.5. Line 214: Change to "immune responses."

Corrected. (line 219)

2.6. Line 216: Change to "different parasite species."

Corrected. (line 221)

2.7. Line 216-217: This point isn't clear. Why would seasonal parasites with short parasitemia be especially likely to encounter long-lasting parasites? Do you mean relative to parasites with short parasitemia encountering other parasites with short parasitemia? (Because long-lasting parasites should also be likely to encounter other long-lasting parasites, right?) If I am understanding your intent correctly, you could rephrase as: "Seasonal parasites with short parasitemia may be especially likely to encounter long-lasting parasites compared to other parasites with short parasitemia. The implications of encounter between parasites with different *infection durations (*or "life histories"?) is worth future inquiry." (Note also that I suggest deleting the "Another interesting direction is related to the interaction between..." prelude to the sentence.)

We thank the reviewer for this comment. We removed this point from the revised version.

2.8. Line 220: You can delete the word "probably."

Deleted.

2.9. Line 222: Simplify to: "In our seasonal system, temperature plays..."

Done as per reviewer's suggestion. (line 227)

2.10. Line 224: Rephrase as "We found water temperature to be an important predictor..."

Done as per reviewer's suggestion. (line 228)

2.11. Line 227: Citation needed at end of sentence: "...likelihood of infection (citation needed here)."

We thank the reviewer for this comment. We cited Ebert et al. (2000) who demonstrated the effects of spore dose on infection success. (line 231)

2.12. Line 227-228: Is the ambiguity in this sentence ("seem to"...."exceptions have been reported") reflecting methodological differences between studies, differences between parasite species, differences between provenance of particular parasite species (e.g., parasite populations could be locally adapted to different environments, and therefore have different thermal performance responses), or different host species contexts (e.g., Gh x E or Gh x Gp x E interactions)? Please clarify which of those factors contribute to differences in parasite thermal performance alluded to in this sentence.

We thank the reviewer for turning our attention to this ambiguity. We replaced 'seems' with 'generally' and changed the end of the sentence to 'but see' (lines 231-232). We think that this new structure describes better that most studies of *D. magna* parasites development under different temperatures demonstrated our statement, but the one counter example is given in citation 75.

2.13. Line 235: "Host" is misspelled.

Corrected. (line 238)

2.14. Line 239: Change "setup" to simply "set."

Done as per reviewer's suggestion. (line 240)

2.15. Line 247: Can delete "We found that" and start sentence with "The observed prevalence..."

Done as per reviewer's suggestion.

2.16. Line 248: No hyphen needed in null model.

Corrected.

2.17. Line 249: Delete "of" to change to: "various host-parasite systems."

Corrected. (line 261)

2.18. Line 251: what does "assimilate" mean in this instance? Do you mean "acquire"?

The sentence was removed from the revised version.

2.19. Line 252: Change to "non-randomly."

The text is already as suggested by the reviewer. The line separation is due to the software limitation and not due to an accidental space.

2.20. Line 255: Change "ended in" to "yielded" or "resulted in."

Changed to 'resulted in.' line (264)

2.21. Line 256: With the network analysis, are you actually "eluding associations"? Or "resolving" them? Or "accounting for" them?

We thank the reviewer for this comment. We changed 'elude' to 'unravel' which better

describes what we attempted to do (find out if there are some pairwise associations that eluded the null model). (line 267)

2.22. Line 286: Replace "bare" with "bear."

Corrected. (line 313)

Reviewer #3 (Remarks to the Author):

Overall, I appreciate the efforts taken to revise this manuscript. The authors addressed many of my comments.

3.1. However, there is still a disconnect between the focus of the introduction and that of the discussion. The introduction largely focuses on parasite interactions and how they affect host health. Whereas the discussion largely focuses on the temporal nature of the system. I think the discussion could be improved by a paragraph explicitly discussing parasite-parasite competition/interactions in the host and speculation as to why they might not have occurred/been important in this system.

Alternatively, if the authors do not feel like there is enough information from their study and outside sources to address these questions in this system, then the focus of the introduction could be shifted away from parasite interactions and be focused more on the topics covered in the discussion. Such as the temporal nature of this host-parasite system or the neutral theory of community ecology.

We thank the reviewer for this comment. We added a new paragraph to the Discussion dealing with the question "why interactions known from the lab are not reflected in our observations". (272-299)

More minor line-by-line comments are below:

3.2. Line 43: There is an extra space between 'survival' and 'can'.

Corrected.

3.3. Lines 36-46: I appreciate that effort was put into this section but I still think it could use work. Overall, I think this paragraph should be reassessed for flow and purpose. Specifically, this paragraph covers parasite-parasite interactions and the effects of coinfections on host health. I find that the transition between these two topics is a bit confusing and that in some places it's hard to keep track of whether the host or parasite is being discussed. For example, the two sentences on lines 40-42 are discussing damage that the co-infections cause to the host, but the sentence starting with 'the effect of...' abruptly switches to host survival rather than host damage and also discusses parasite competition. Some transition there would be appreciated. Additionally, on line 43 when 'modified transmission rate' is discussed, I believe this would be clearer if it was written 'modified parasite transmission rate'. Finally, the sentence starting on line 44 is particularly confusing. As both hosts and parasites reproduce it would be nice to have some clarity as to which organisms' reproduction is being affected.

We thank the reviewer for this comment. We changed the wording of this part and now the text is more uniform. Specifically, we replaced 'competition' with 'coinfection' and removed

host survival, while connecting the sentence better to the previous one. These changes keep the same meaning and intention of the text, while using more unified terminology. (lines 42-44)

3.4. Line 57: I'm not sure 'correspondingly' is the correct word to use in this context. I would remove.

Deleted.

3.5. Line 72: Suggest changing 'several studies examined' to 'several studies have examined'.

Corrected. (line 72)

3.6 Line 75: This sentence is confusing. I think because 'develop an infection' is a phase usually associated with a host and not a parasite. Suggested rewording: 'Within-host competition between differently virulent parasites can limit the ability of the less virulent parasite to successfully infect the host'.

We thank the reviewer for this comment. We carried out the reviewer's suggestion. (lines 75-76)

3.7. Line 78: Same issue with 'form an infection' as described above.

We reworded to 'infect its host'. (line 78)

3.8. Line 89: Suggest changing 'combining' to 'combined'.

Corrected. (line 89)

3.9. Line 157: Suggest change from "that affect the coinfection rate, Simpson's" to "that affect coinfection rate: Simpson's".

Corrected. (line 163)

3.10. Line 194-197: Long sentence. Consider breaking it up.

We thank the reviewer for this comment. We split the sentence and moved the last part to another place in the paragraph to improve the flow. (line 199 and lines 206-208)

3.11. Line 214: Suggest changing 'response' to 'responses'.

Corrected. (line 219)

3.12. Line 215: I'm not sure what the purpose of the phrase 'though overlooked' is in this sentence. Are you saying that nobody has studied coinfections in this system?

We thank the reviewer for this comment. We reworded to 'not tested'. (line 220)

3.13. Your discussion devotes 4 paragraphs (over 2 pages of text) to seasonality. While I agree this is an important part of your study and should be well addressed, this section could be reduced. As an example, I was fond of the paragraph in lines 220-231. It was concise and the arguments/connection to your own work was clear. Additionally, perhaps if this seasonality section was reduced, sometime could be

spent on how highly temporally dependent host-parasite systems might generate different within-host interactions than would be seen in longer-term systems.

We appreciate the reviewer's comment. We attempted to tone down the "presence" of temporality in the paper by reducing the length of one paragraph (lines 209-224) and adding more text to the part devoted to the absence of interactions (lines 272-299). Identifying the temporal pattern of coinfection was one of our main objectives and as such we covered it thoroughly in the Discussion. Our discussion of the topic, in our point of view, comprised of two different angles. First, by explaining how the dynamics of different parasites relate to the dynamics of coinfection (hence the paragraph discussing the pattern directly and the one connecting our ideas and results to other systems). Second, by examining environmental aspects that may cause the observed patterns (hence the temperature and habitat instability paragraph). While comparing short- and long-term systems will be fascinating (we do imply to something similar in the end of the habitat instability paragraph), it must have come at the expense of at least one of the ideas we chose to discuss and therefore we prefer to keep the points that we believe are interesting and important to discuss, rather than replacing them with others. Moreover, we think that such a comparison deserves a paper of its own accompanied with a comparative study, since we believe that several lines of inquiry are needed for that.

3.14. Line 235: I am not sure what 'hist density' is. Host density?

Corrected. (line 238)

3.15. Line 248: When you say 'null models were used' this suggests that you ran these models. However, as you have citations, I believe this phrase refers to models that other people have run. Suggest rewording to 'null models have been used' to clarify that this is in reference to previous work.

We thank the reviewer for this comment and applied the reviewer's suggestion. (line 260)

3.16. Line 258: What does 'e.g.' refer to? Was there supposed to be an example here? If you're referring to the citation, I think you can just remove the e.g.

It seems that the reviewer referred to the phrase in line 252 'even when studying the whole community, e.g.,'. We removed the "e.g." and added a citation.

3.17. Line 257: Suggest changing to 'revealed a strong correlation'.

Corrected. (line 267)

3.18. Line 285: If these parasites 'probably bare more costs' it would be nice to write why. If it is not probable or challenging to speculate on why you could change this to 'may bare more costs'.

We thank the reviewer for this comment. The word 'probably' was replaced by 'may'. (line 313)

3.19. Line 349: I think the figure you label here as figure s1 is now figure s12.

We corrected the name of the figure which is now S15. (line 375)

3.20. Line 381-382: Consider rephrasing/reassessing grammar in the sentence starting with "we calculated the...". I'm not following.

We thank the reviewer for this comment. We rephrased this sentence as follow, "To estimate the strength of the edges, we calculated their weights as the sum of hosts coinfecting by each pair". We think it is better connected to the previous sentence and easier to follow. (lines 407-408)

3.21. Table 1: I appreciate the inclusion of this table but it is really clunky. It looks like some species names are bolded and some are not. The word 'amoeba' in the group column appears to be a different font than the rest of the table. The figure legend is missing an ending period. I would also recommend widening the 'changes under coinfections' column to reduce table size.

1) None of the species names are bolded in our original files. All the names are italicized except for descriptive names that are not scientific, e.g., 'Unknown microsporidium spp.'. We hope that whatever caused some of the names to appear as bolded in the file sent to the reviewers will be fixed with the upload of the revised files.

2) We thank the reviewer for noticing the different fonts, this was fixed in the revised version.

3) We attempted to adjust the columns as much as possible under the limitations of Microsoft Word. While hoping it improves the visibility of the table in the submitted files, we believe that final adjustment will be made together with the journal's copyediting team, if the paper is accepted.

3.22. Supplementary data: This needs another look over for spelling. On the 'List of GLMM final' sheet oxygen is spelled wrong every time and diversity is spelled wrong in model 6. However, I appreciate all the changes made to this. It is much clearer and matches the manuscript.

We thank the reviewer for this comment. We corrected all spelling mistakes in the file.

3.23. Figures S10 & S11. I think changing the line type vs. dot shape (as you do in Figure 2) for the different parasite groups may generate clearer definition between the groups. If it is still challenging to see differences in line types (dashed vs. not) when the lines overlap, perhaps you can try both dots and lines would be the clearest.

We thank the reviewer for this comment. We changed the line type of gut parasites in these figures to dash-line. In addition we replaced the diamond marker with x markers that are more distinguishable from circles.

REVIEWERS' COMMENTS:

Reviewer #1 (Remarks to the Author):

This is the third time I have read the manuscript by Halle et al. where *Daphnia magna* populations were sampled through time for parasites to see how parasite and coinfection prevalence's develop, which parasites form coinfections, and which other conditions affect coinfection prevalence. In this round of revisions, the authors present new analyses as argumentation for not including abiotic variables in the host density models and present plots of host density to respond to my previous comments. The discussion has been reordered for better flow and a section on co-infections has been included. However, the discussion is long at 6 pages and 10 paragraphs and there is a full paragraph that is not discussing the results of this study.

1. In the previous round of comments, I wrote that the paragraph L209-223 is discussing results of other studies. It is also not clear why the selected examples would be so important that they should be explored in such detail. The function of the section has not become clearer after revising. This paragraph does not include any discussion of the results of this study and seems to repeat the topic of the previous paragraph. Seasonality is already discussed in the previous paragraph. One paragraph would be sufficient especially since the discussion is long at 6 pages and 10 paragraphs and another reviewer also pointed out that the heavy focus on Seasonality seems out of balance.

Typos and other small things:

2. In the previous round I suggested changing the sentence L127 into "We found coinfection in 33.28% of the infected *D. magna* individuals." but it has accidentally been changed into "We found coinfecting *D. magna* in 33.28% of the infected *D. magna* individuals."
3. L142 extra hyphen in "pre-liminary"
4. L171-172 the sentence is missing a verb
5. L180 "turned as insignificant" -> "were found to be insignificant"
6. L180 extra hyphen in "pre-liminary"
7. L199 "disrupt" should be "disrupted"
8. L265 Please add "null" before "model" to distinguish between the different models
9. L271 typo in "parasite" which should be "parasites"
10. L271-272 "were observed" sounds like you observed it in this study. Replacing with "have been observed" and adding the references would help distinguish current and previously published results
11. The beginning of the new paragraph on coinfections (L271-298) is a little confusing in terms of which analysis is referred to. It would help to 1) L272 use "network analysis" instead of "network analysis" which would help the readers to immediately think of the analysis out of the many analysis presented, and 2) L274 "footprint on the prevalence of coinfection or of specific species." makes it sound like this is not about the network analysis
12. L284 unnecessary "very", suggest using "highest" instead of very high
13. L288 has a comma in "These two examples, highlight..."
14. L301 is missing a verb
15. L449 has both an extra comma and a typo in "Since, most of the parameter measured..." . Should be "parameters"
16. L450 typo in 'corelated', should be 'correlated'

Reviewer #2 (Remarks to the Author):

This is my third time reviewing this manuscript. I have thoroughly read the other reviewers' comments and author responses, and appreciate the thoughtful effort that both the reviewers and authors have

put into improving this manuscript.

Here, I reply directly only to the authors' responses to my own previous comments, focusing on those comments that were not simply grammatical/spelling issues (I was reviewer #2 in the previous round):

2.1. Line 44 ("...on their reproduction"): replace "their" with either "host" or "parasite". Currently unclear which is being referred to. Effects on either host or parasite reproduction could affect disease transmission.

Corrected. (line 44).

Reviewer: Thank you, this is much clearer now.

2.4. Line 206-207: "Similar to our findings, the two parasites had overlapping prevalence pattern in both host and vectors and coinfections were present." Change to "patterns." Also, did that study indicate how prevalent coinfections were? That coinfections were "present" doesn't give us much information, so perhaps you could tell us if coinfections were common or not.

We thank the reviewer for this comment. We corrected 'pattern' to 'patterns' (lines 212). As for the additional info regarding coinfection in this study, the percentage of coinfections in the robins is not stated in the paper and for the mosquitos the probability of having one parasite is higher if the other parasite is present, but data of coinfections in the samples is absent. So unfortunately stating anything more indicative about coinfections from this study would be speculative.

Reviewer: Ok, thank you for explaining.

2.7. Line 216-217: This point isn't clear. Why would seasonal parasites with short parasitemia be especially likely to encounter long-lasting parasites? Do you mean relative to parasites with short parasitemia encountering other parasites with short parasitemia? (Because long-lasting parasites should also be likely to encounter other long-lasting parasites, right?) If I am understanding your intent correctly, you could rephrase as: "Seasonal parasites with short parasitemia may be especially likely to encounter long-lasting parasites compared to other parasites with short parasitemia. The implications of encounter between parasites with different *infection durations (*or "life histories"?) is worth future inquiry." (Note also that I suggest deleting the "Another interesting direction is related to the interaction between..." prelude to the sentence.)

We thank the reviewer for this comment. We removed this point from the revised version.

Reviewer: Ok.

2.11. Line 227: Citation needed at end of sentence: "...likelihood of infection (citation needed here)."

We thank the reviewer for this comment. We cited Ebert et al. (2000) who demonstrated the effects of spore dose on infection success. (line 231)

Reviewer: Ok.

2.12. Line 227-228: Is the ambiguity in this sentence ("seem to"...."exceptions have been reported") reflecting methodological differences between studies, differences between parasite species, differences between provenance of particular parasite species (e.g., parasite populations could be

locally adapted to different environments, and therefore have different thermal performance responses), or different host species contexts (e.g., Gh x E or Gh x Gp x E interactions)? Please clarify which of those factors contribute to differences in parasite thermal performance alluded to in this sentence.

We thank the reviewer for turning our attention to this ambiguity. We replaced 'seems' with 'generally' and changed the end of the sentence to 'but see' (lines 231-232). We think that this new structure describes better that most studies of *D. magna* parasites development under different temperatures demonstrated our statement, but the one counter example is given in citation 75.

Reviewer: Ok.

2.21. Line 256: With the network analysis, are you actually "eluding associations"? Or "resolving" them? Or "accounting for" them?

We thank the reviewer for this comment. We changed 'elude' to 'unravel' which better describes what we attempted to do (find out if there are some pairwise associations that eluded the null model). (line 267)

Reviewer: I agree that "unravel" is a good word choice here.

Reviewer #3 (Remarks to the Author):

I am excited to see that the authors took the time to rewrite and remove portions of the discussion to better tie their results in with both their introduction and with the boarder body of coinfection literature. I have included some minor line-by-line comments below that are primarily grammatical/formatting.

Line 84: There is an extra space between 'observed' and the citation.

Line 178: I not sure what this sentence is saying. Consider rephrasing.

Line 180: Please change 'pre-liminary' to 'preliminary'.

Line 199: Change 'disrupt' to 'disrupted'.

Line 207: Unclear phrasing potentially due to grammatical issues. I suggest changing it to something like: '...the importance of the prevalence of multiple parasites...'

Lines 209-211: This sentence comes across as defensive. I would suggest something like: "Strong temporal patterns of coinfections are common due to the commonness of temporal and seasonal patterns in natural populations, including those of parasites."

Line 212: There appears to be an extra space between 'patterns' and 'of'.

Line 215: There appears to be an extra space between 'study' and the citation.

Line 222: 'Instrument' means a physical tool to me, not a task or concept. I would replace this word.

Lines 247-251: I would remove the opening sentence of this paragraph and combine the following two

sentences like so: "In the field, mechanisms and processes other than parasite interactions (e.g. genetic interactions among hosts and parasites or environmental effects) can reduce or even mask the effects of parasite-parasite interactions found in laboratory settings." I also think this paragraph would be nice if it was right before the paragraph that starts on line 271 because these two paragraphs are highly related.

Line 273: Footprint is one word in US English at least.

Lines 271-297: Thank you for adding this paragraph. I think it adds some nice context.

Supplemental figures S10 & S11. These figures will always be busy, but I like what you did with the colors and shapes/line types. It makes it possible for a reader to find a specific pattern if they're interested.

We thank the reviewers for their positive feedback and suggestions on our last version. We corrected all the grammatical, technical and minor wording issues pointed by the reviewers. Bigger comments made by R1 and R3 were also addressed as followed:

R1

In the previous round of comments, I wrote that the paragraph L209-223 is discussing results of other studies. It is also not clear why the selected examples would be so important that they should be explored in such detail. The function of the section has not become clearer after revising. This paragraph does not include any discussion of the results of this study and seems to repeat the topic of the previous paragraph. Seasonality is already discussed in the previous paragraph. One paragraph would be sufficient especially since the discussion is long at 6 pages and 10 paragraphs and another reviewer also pointed out that the heavy focus on Seasonality seems out of balance.

We made additional changes to this paragraph. Specifically, we reduced its length and changed the opening sentence of the paragraph to improve the flow from the previous paragraph. We believe that now this part is less intensive and yet still contributing a broader context to the relevancy of our results and ideas. (lines 209-219)

The beginning of the new paragraph on coinfections (L271-298) is a little confusing in terms of which analysis is referred to. It would help to 1) L272 use "network analysis" instead of "network analysis" which would help the readers to immediately think of the analysis out of the many analysis presented, and 2) L274 "footprint on the prevalence of coinfection or of specific species." makes it sound like this is not about the network analysis

We changed the words "our analysis" in the opening sentences to "our null model followed by network analysis" (line 268). This specifies the two analysis that we refer to in the beginning of this paragraph and should ease on the readers as R1 suggested.

R3

Line 178: I not sure what this sentence is saying. Consider rephrasing.

We changed the sentence to " Seasonal changes in host density in each pond along with the prevalence of coinfections and infections by a single parasite species are presented in Figure S14". Now it is clearer and more in context with jargon of the paper.

This sentence comes across as defensive. I would suggest something like: "Strong temporal patterns of coinfections are common due to the commonness of temporal and seasonal patterns in natural populations, including those of parasites."

We agreed with this comment but decided on a different opening sentence as part of the changes the we made following the first comment of R1. The new version is "The relationship between overlapping parasite prevalence and coinfections should be relevant for other systems" (lines 209-210)

Lines 247-251: I would remove the opening sentence of this paragraph and combine the following two sentences like so: "In the field, mechanisms and processes other than parasite interactions (e.g. genetic interactions among hosts and parasites or environmental effects) can reduce or even mask the effects of parasite-parasite interactions found in laboratory settings." I also think this paragraph would be nice if it was right before the paragraph that starts on line 271 because these two paragraphs are highly related.

We took the reviewer suggestion both for the opening sentence (lines 255-257) and the location of this paragraph (lines 255-256). We would like to mention that R3's comment regarding the location of the paragraph is contradicting to a comment made by R1 in the previous round of revision and we found ourselves in better agreement with R3 point of view.

All the changes made in this final version are highlighted in yellow in the Halle et al. 1.4.4 track changes.docx.

We would like to thank you again for your time and kindness during the revision process.